# Phase diagram for strong-coupling Bose polarons

Arthur Christianen[1,2]*, J. Ignacio Cirac[1,2] and Richard Schmidt[3,†]

**1** Max-Planck-Institut für Quantenoptik, Hans-Kopfermann-Str. 1, D-85748 Garching, Germany
**2** Munich Center for Quantum Science and Technology (MCQST), Schellingstraße 4, D-80799 Munich, Germany
**3** Institute for Theoretical Physics, Heidelberg University, Philosophenweg 16, 69120 Heidelberg, Germany
* arthur.christianen@mpq.mpg.de, † richard.schmidt@thphys.uni–heidelberg.de

August 1, 2023

## Abstract

**Important properties of complex quantum many-body systems and their phase diagrams can often already be inferred from the impurity limit. The Bose polaron problem describing an impurity atom immersed in a Bose-Einstein condensate is a paradigmatic example. The interplay between the impurity-mediated attraction between the bosons and their intrinsic repulsion makes this model rich and interesting, but also complex to describe theoretically. To tackle this challenge, we develop a quantum chemistry-inspired computational technique and compare two variational methods that fully include both the boson-impurity and interboson interactions. We find one regime where the impurity-mediated interactions overcome the repulsion between the bosons, so that a sweep of the boson-impurity interaction strength leads to an instability of the polaron due to the formation of many-body clusters. If instead the interboson interactions dominate, the impurity will experience a crossover from a polaron into a few-body bound state. We achieve a unified understanding incorporating both of these regimes and show that they are experimentally accessible. Moreover, we develop an analytical model that allows us to interpret these phenomena in the Landau framework of phase transitions, revealing a deep connection of the Bose polaron model to both few- and many-body physics.**

# 1   Introduction

Understanding the properties of complex quantum many-body systems is one of the major challenges in modern physics [1,2]. Gases of ultracold atoms provide a unique playground to study such systems in a controlled environment [3–7]. From a theoretical perspective, a special feature of these systems is the practically complete understanding of the microscopic interactions between the atoms [8]. Moreover, the universality of ultracold scattering enables the translation of these detailed models into simple effective Hamiltonians, which can still be used to describe experiments on a quantitative level. This has led to great synergy between theory and experiments and a significantly improved understanding of paradigmatic systems such as the Fermi-Hubbard model [9–16], the unitary Bose-Einstein condensate (BEC) [17–24], and the crossover from a BEC to a BCS (Bardeen-Cooper-Schrieffer) state of paired fermions [25–30]. Nevertheless, even though the theoretical models and Hamiltonians seem simple at first sight, important aspects of these systems are still not understood.

Particularly interesting systems are ultracold mixtures of different species of bosons and/or fermions. For example, boson-mediated interactions between fermions are a classic and important mechanism of superconductivity [31]. With cold atoms, also the opposite scenario of fermion-mediated interactions in a BEC has been studied [32,33]. Mixtures of bosons on the other hand, can form quantum droplets [34–39]. Other than being of fundamental interest, ultracold mixtures are also a common starting point for creating ultracold molecules [40–44]. A good understanding of ultracold mixtures can therefore also be harnessed to create ultracold molecules more efficiently [45].

The rich properties and the variety of phenomena that make these ultracold mixtures so

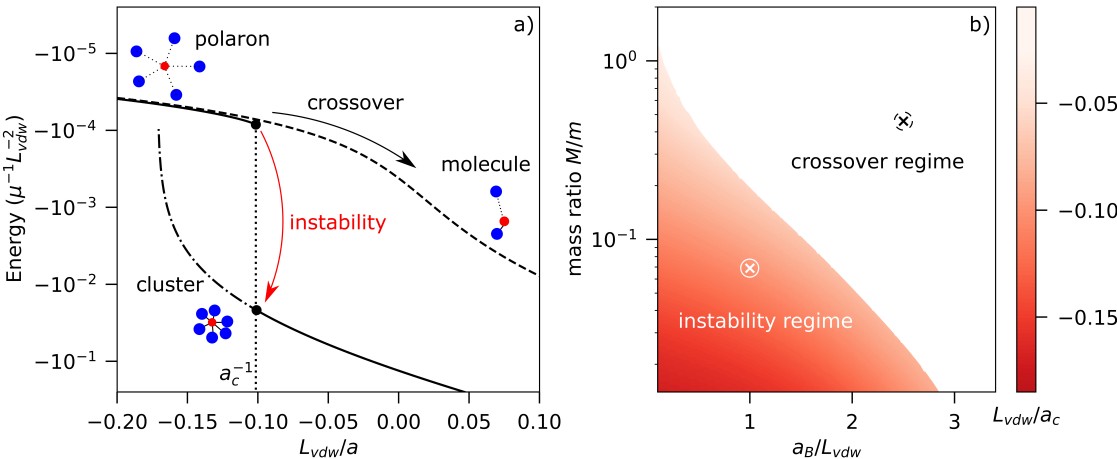

Figure 1: a) Energy of the Bose polaron as a function of the inverse impurity-boson scattering length $1/a$ in the regime where it undergoes an instability (bold) or crossover (dashed) as a function of the interaction strength. In the instability regime, the polaron state becomes unstable at the scattering length indicated by the dot. At this point the polaron decays into a cluster. The energy of the cluster before the instability is drawn with the dash-dotted line (only shown in the regime where it is lower than the polaron energy). The bold and dash-dotted lines are computed using a Gaussian-state Ansatz, and the dashed line using a double-excitation Ansatz (see Sec. 2). b) Stability diagram indicating whether an instability or crossover will occur as a function of the impurity-boson mass ratio $M/m$ and the interboson repulsion scattering length $a_B$. In the lower left part of the diagram, the scattering length of the instability $a_c$ is indicated by the colormap. The crosses indicate the parameters corresponding to the lines in subfigure a). For both a) and b) scales characterizing the interaction ranges of the boson-impurity and boson-boson potential, $L_g = L_U = 2L_{vdw}$ are chosen to be equal (see Sec. 2.3). For the density of the BEC we have chosen a typical value of $n_0 = 10^{-5}L_g^{-3}$ (approximately $10^{14}$ cm$^{-3}$).

interesting have the immediate consequence that the development of a unified theoretical picture or a global phase diagram is extremely challenging [46]. A successful approach to achieve essential insights has been to start from the quantum impurity limit, of a single particle of one species immersed in a sea of the other. Often, key signatures of the many-body properties of the full mixture can already be recognized in this well-defined limit.

A good example is the problem of the Fermi polaron [47–52], an impurity immersed in a Fermi sea. As a function of interaction strength, this model shows a transition from a polaronic to a molecular ground state, which is a precursor of the BEC to BCS crossover. Furthermore, this model shows an interesting connection to the elusive Fulde-Ferell-Larkin-Ovchinnikov phase of superconductivity [53–56].

The Bose polaron problem [57–62] of an impurity immersed in a BEC has proven to be more challenging. We focus on the case of a charge-neutral impurity, but also the case of an ionic impurity has drawn recent theoretical interest [63–65]. Despite much effort, a unified understanding is lacking of what happens to the impurity as a function of interaction strength. The complications which arise here are the large number of excitations around the impurity, the importance of higher-order correlations, and the interactions between bosons from the BEC. In three dimensions[1], no theoretical approach has so far captured all of these aspects

---

[1]In contrast, in the 1D case even analytical expressions can be obtained that describe the strong-coupling Bose polaron relatively well [66].

simultaneously.

In Fig. 1a) we illustrate how the character of the attractive Bose polaron is predicted to change as the boson-impurity interactions are swept across a Feshbach resonance. Specifically, we show the energy of the polaron as a function of the inverse scattering length. State-of-the-art theories envision two different scenarios. In the first scenario, the polaron experiences a smooth crossover into a small molecule, such as a dimer, trimer, or tetramer [67–69]. In the second scenario, an instability is predicted, in the form of a collapse of the BEC triggered by attractive impurity-mediated interactions [70,71]. The result is the decay of the polaron into a multi-particle bound cluster. These scenarios are qualitatively different, and so far they have not been captured together in a single theoretical framework.

In this work we achieve a unified picture of the properties of the attractive Bose polaron as a function of the BEC density, impurity mass, the boson-impurity and boson-boson interaction strengths. We show in Fig. 1b), that, depending on the system, both of the scenarios in Fig. 1a) can occur. The largest part of the parameter space corresponds to the crossover regime. The crossover occurs if the impurity is relatively heavy or similar in mass to the bosons of the BEC, or if the interboson scattering length $a_B$ is significantly larger than the van der Waals length. This is the regime where most experiments would naturally be realized. However, the instability predicted in Refs. [70,71] persists in presence of modest interboson repulsion for light impurities, in a regime which is also experimentally achievable.

To arrive at the unified picture of Bose polarons, it is crucial to explicitly incorporate the interactions of the bosons in the BEC into the model, and to compare the variational Gaussian-state and double-excitation approaches. Importantly, both of these methods also incorporate the impurity-mediated interactions between bosons and the Efimov effect. To facilitate efficient numerical implementation, we parameterize the variational wave functions in terms of quantum chemistry-inspired Gaussian basis sets.

Finally, using a simple Gaussian wave function, we capture the qualitative behaviour of the Bose polaron in an analytical model. This links the phenomena we see in the Bose polaron model to the paradigmatic Landau model of first- and second-order phase transitions [72]. In fact, we can interpret the instability-to-crossover physics as an analog of the typical liquid-to-gas transition, but appearing at zero temperature.

The structure of this paper is as follows. In section 2 we introduce the model and the theoretical methods, and in section 3 we give an overview of the theory background. Then we demonstrate how we include the renormalized interboson interactions with Gaussian states in section 4. In section 5, we describe our numerical results showing the transition from the instability to the crossover regime. The analytical model that captures this behaviour is presented in section 6. Finally, we conclude our work in section 7 and we provide an outlook on future directions and interesting avenues to pursue.

## 2 Theoretical and computational methods

### 2.1 Hamiltonian and variational methods

We consider the problem of a mobile impurity of mass $M$ in a homogeneous BEC of bosons with mass $m$ and chemical potential $\mu_B$. We denote the impurity-boson interaction potential by $V_{IB}$ and the boson-boson interaction potential by $V_{BB}$. The interactions are treated within a single-channel model, assuming the boson-impurity scattering length is tuned close to a broad Feshbach resonance. We treat the impurity in first quantization with quadrature operators $\hat{\boldsymbol{R}}$ and $\hat{\boldsymbol{P}}$, and the bosons in second quantization with creation and annihilation operators $\hat{b}_{\boldsymbol{k}}^{\dagger}$ and $\hat{b}_{\boldsymbol{k}}$, respectively. We set $\hbar = 1$. This gives the following Hamiltonian

$$\hat{\mathcal{H}}_0 = -E_{bg} + \int_k \Big(\frac{k^2}{2m} - \mu_B\Big)\hat{b}_k^\dagger \hat{b}_k + \frac{\hat{P}^2}{2M} + \int_r V_{IB}(r - \hat{R})\hat{b}_r^\dagger \hat{b}_r + \frac{1}{2}\int_{r'}\int_r V_{BB}(r' - r)\hat{b}_{r'}^\dagger \hat{b}_r^\dagger \hat{b}_{r'}\hat{b}_r.$$
(1)

Here $E_{bg}$ is the energy of the background BEC without the impurity. We denote $\int_r = \int d^3 r$ and $\int_k = \frac{1}{(2\pi)^3}\int d^3 k$. The chemical potential is set to the mean field value $\mu_B = n_0 U_B$, where $n_0$ is the density of the BEC and $U_B$ the coupling constant on the level of the Born approximation. To approximate the ground state of the Hamiltonian we consider a variational Ansatz of the type

$$|\psi\rangle = \hat{U}_{n_0}\hat{U}_{LLP}\hat{A}(x)|0\rangle.$$
(2)

Here the unitary $\hat{U}_{n_0}$ displaces the background condensate described by a coherent state and

$$\hat{U}_{LLP} = \exp(-i\hat{R}\int_k k\, \hat{b}_k^\dagger \hat{b}_k)$$
(3)

performs the Lee-Low-Pines transformation to transform to the reference frame of the impurity [73,74]. The operator $\hat{A}(x)$ is dependent on variational parameters $x$, which are optimized to minimize the energy; $|0\rangle$ is the bosonic vacuum state with the impurity at the center of the comoving frame. In the following we set the total momentum of the system to zero, which yields the following transformed Hamiltonian.

$$\begin{aligned}
\hat{\mathcal{H}} &= \hat{U}_{LLP}^\dagger \hat{U}_{n_0}^\dagger \hat{\mathcal{H}}_0 \hat{U}_{n_0}\hat{U}_{LLP} \\
&= \int_k \frac{k^2}{2\mu_r}\hat{b}_k^\dagger \hat{b}_k + \frac{1}{2M}\int_{k'}\int_k k'\cdot k\, \hat{b}_{k'}^\dagger \hat{b}_k^\dagger \hat{b}_{k'}\hat{b}_k + \int_r V_{IB}(r)(\hat{b}_r^\dagger + \sqrt{n_0})(\hat{b}_r + \sqrt{n_0}) \\
&+ \int_{r'}\int_r V_{BB}(r'-r)\Big[\frac{n_0}{2}(2\hat{b}_{r'}^\dagger \hat{b}_r + \hat{b}_{r'}^\dagger \hat{b}_r^\dagger + \hat{b}_{r'}\hat{b}_r) + \sqrt{n_0}(\hat{b}_{r'}^\dagger \hat{b}_r^\dagger \hat{b}_r + \hat{b}_r^\dagger \hat{b}_{r'}\hat{b}_r) \\
&\hspace{9cm} + \frac{1}{2}\hat{b}_{r'}^\dagger \hat{b}_r^\dagger \hat{b}_{r'}\hat{b}_r\Big].
\end{aligned}$$
(4)

The second term in the transformed Hamiltonian originates from the Lee-Low-Pines transformation applied to the impurity momentum operator. It is this term which gives rise to impurity-mediated interactions between the bosons [70,71], crucial for the description of the Efimov effect [75].

In this work we will compare two types of variational Ansätze. The first is a Gaussian-state (GS) Ansatz

$$\hat{A}_{\mathrm{GS}}[\mathcal{N}, \phi(k), \xi(k, k')] = \mathcal{N}\exp\Big[\int_k (\phi(k)\hat{b}_k^\dagger - \phi^*(k)\hat{b}_k)\Big]\exp\Big[\frac{1}{2}\int_k \int_{k'}\hat{b}_k^\dagger \xi(k, k')\hat{b}_{k'}^\dagger\Big].$$
(5)

Here the subclass where $\xi = 0$ is referred to as a coherent-state (CS) Ansatz. Importantly, with Gaussian states one can compute expectation values with Wick's theorem [76], simplifying calculations.

The second wave function is a double-excitation (DE) Ansatz, given by

$$\hat{A}_{\mathrm{DE}}[\beta_0, \beta(k), \alpha(k, k')] = \beta_0 + \int_k \beta(k)\hat{b}_k^\dagger + \frac{1}{\sqrt{2}}\int_k \int_{k'}\alpha(k, k')\hat{b}_k^\dagger \hat{b}_{k'}^\dagger.$$
(6)

For the double-excitation Ansatz, the case where $\alpha = 0$ is often called the Chevy Ansatz, after Chevy, who first introduced an Ansatz of this kind for the Fermi polaron problem [47].

## 2.2 Basis set and computations

We parameterize our wave functions in terms of a Gaussian basis set. This approach is inspired by quantum chemistry where the use of Gaussian basis functions is common practice [77]. Concretely, in the Gaussian-state case, this parameterization corresponds to

$$\phi(\mathbf{k}) = \sum_i \phi_i \chi_{00}(\sigma_i^{(\phi)}, \mathbf{k}), \tag{7}$$

$$\xi(\mathbf{k}, \mathbf{k}') = \sum_{lij} \sum_{m=-l}^{l} (-1)^m \xi_{ij}^{(l)} \chi_{lm}(\sigma_i^{(\xi,l)}, \mathbf{k}) \chi_{l-m}(\sigma_j^{(\xi,l)}, \mathbf{k}). \tag{8}$$

where the functions $\chi_{lm}(\sigma, \mathbf{k})$ are spherical Gaussian basis functions,

$$\chi_{lm}(\sigma, \mathbf{k}) = (2\pi)^{3/2} Y_{lm}(\theta, \phi) i^{-l} k^l \exp(-\sigma k^2), \tag{9}$$

with spherical harmonics $Y_{lm}(\theta, \phi)$. In Appendix A we show how to compute expectation values with the Gaussian states and Gaussian basis functions.

Since the polaron cloud has a smooth shape and is localized around the impurity, this approach requires much fewer variational parameters than in our previous approach in Refs. [70, 71], where the wave functions were parameterized by simply discretizing $\phi$ and $\xi$ in a spherical wave basis. Gaussian basis functions are chosen over other types of basis function which might more closely resemble shape of the polaron cloud, because integrals over Gaussian functions give simple analytical expressions. In particular, the matrix elements of the interboson interactions generally take a complicated form, whereas for Gaussian basis functions they can still be computed analytically, at least for Gaussian potentials.

For the calculations one can either choose to keep the exponents of the Gaussian basis functions fixed, or to also treat them as variational parameters. Here we leave them fixed. The size of the smallest $\sigma$ is determined by the range of the potential, and the size of the largest $\sigma$ by the extent of the polaron cloud. Since these length scales are orders of magnitude different, we choose the values of $\sigma$ to be spaced logarithmically. The spacing of $\sigma$ is then chosen by ensuring convergence of the parameters of interest. For varying calculations we typically use between five and twenty values of $\sigma$ per angular momentum mode, depending on the variational method, the observable, and the desired convergence.

In the Gaussian-state case, we optimize the variational parameters $\phi$ and $\xi$ by using imaginary time evolution, for which the equations are derived in Appendix B. Since the problem is spherically symmetric, we can restrict ourselves to only the zero angular momentum modes for $\phi$. For $\xi$, the two created particles always need to have opposite angular momenta. We solve these equations numerically with a solver based on backward-differentiation formulas [78, 79], which greatly outperforms standard Runge-Kutta methods for this problem due to the stiffness of the differential equations. The stiffness originates from the interplay of the vastly different length scales of the range of the potential and the healing length of the BEC. We find that using the Gaussian basis set, qualitative and near-quantitative results can already be retrieved with a relatively small number of parameters. However, in the regimes with the strongest correlations the stiffness of the non-linear equations of motion can lead to problems reaching strict convergence when increasing the number of parameters [2].

For the double-excitation Ansatz, most computation steps proceed analogously. Here one can also derive equations of motion for imaginary time evolution. However, opposed to the

---

[2] In our implementation, it is the stiffness of the differential equation rather than the direct scaling with the number of parameters which limits the computational cost. When too many basis functions are included, the basis set comes close to creating linear dependencies. This slows down the numerical optimization and it sometimes leads to the solver getting stuck. In the regime of large repulsion, where the correlations are most important, this limits the convergence of our parameters of study to about five percent.

Gaussian-state case, the equations of motion are linear and can be solved much more efficiently by direct diagonalization. Here stiffness is thus less of an issue, and more rigorous convergence can be reached.

## 2.3 Interaction potentials

We model the impurity-boson and boson-boson interactions using Gaussian potentials

$$V_{IB}(\boldsymbol{r}) = \frac{g}{2L_g^2} \exp\left(-\frac{r^2}{L_g^2}\right), \tag{10}$$

$$V_{BB}(\boldsymbol{r}) = \frac{U}{2L_U^2} \exp\left(-\frac{r^2}{L_U^2}\right). \tag{11}$$

Here $L_g$ and $L_U$ set the ranges of the potentials and $g$ and $U$ set the coupling strengths. The matrix elements over these interactions potentials and the spherical Gaussian basis functions are computed analytically and given in Appendix D.

We fix the coupling strengths $g$ and $U$ to give us the desired scattering lengths $a$ and $a_B$, respectively. The scattering lengths corresponding to the Gaussian potentials can simply be determined by solving the two-body problem, or using the simple formulas from Ref. [80]. There is no unique choice of $U$ and $g$, but we take $U > 0$ and for $g$ we take the smallest negative value that reproduces the desired scattering length.

The range of the boson-impurity Gaussian potential can be related to the range of the typical cold-atom van der Waals potential via the effective range $r_{eff}$. We do this at unitarity $a \to \infty$. There, $r_{eff} \approx 1.4L_g$ for the Gaussian potential, whereas for a van der Waals potential $r_{eff} \approx 2.8L_{vdw}$ [8, 81], meaning that $L_g \approx 2L_{vdw}$. For modest positive scattering lengths, the results are less universal, and the repulsive interboson Gaussian potential we use here can generally not reproduce the effective range of a van der Waals potential. By fixing the absolute range relative to $L_g$, however, we believe we still obtain representative results. Note that having finite-range interactions is crucial for the description of the Efimov effect, since the range of the interactions sets the three-body parameter and therefore the scattering length of the first Efimov resonance $a_-$ [75, 82–84].

## 3 Current theoretical status

By now it has been firmly established that the Efimov effect plays an important role in the Bose-polaron problem [67–71]. The Efimov effect has many interesting features [75, 85], but most important for this work is that it is a cooperative binding effect. One speaks of cooperative binding if the ability of particles to bind increases with the number of particles in a bound state. In the case of the Efimov effect for example, three-body bound states can be bound even if the potential is too shallow to support two-body binding. In fact, the cooperative binding of the Efimov effect also persists for more than three particles, both in the homonuclear [86, 87] and the heteronuclear case [70].

In the quantum impurity case, the Efimov effect arises from an effective impurity-mediated interaction between the bosons. The cooperative binding effect comes into play in the following way: The more bosons bind to the impurity, the smaller the impurity kinetic energy per boson becomes. As a result, the bosons can use the attractive boson-impurity interaction more efficiently. In the Hamiltonian of Eq. (4), the second term is responsible for these impurity-mediated interactions, and it originates from carrying out the Lee-Low-Pines transformation. Since the prefactor of this term is $1/M$, one immediately notices that for the heteronuclear

Efimov effect, the mass of the impurity is of crucial importance. The heavier the impurity is, the weaker the mediated interactions are and the more the Efimov effect is hence suppressed.

The exact role the Efimov effect plays in the Bose-polaron framework is still open to debate. Both the crossover from a polaron into an Efimov state [67–69] and a complete polaronic instability caused by the Efimov effect [70, 71] have been predicted.

When a variational Ansatz of the double (or triple) excitation type is used [67,69,88] (or a similar method with an intrinsically limited number of excitations [68]), a crossover behaviour is found by construction. The number of excitations is simply not large enough to describe the formation of bound clusters containing many particles. These results were corroborated with quantum Monte Carlo studies, for equal mass or heavy impurities [88, 89]. In these studies either interboson repulsion or an effective three-body repulsion originating from the use of a two-channel model [67,69,88] prevent the build-up of many excitations on the polaron.

If an arbitrary number of excitations is allowed, a divergence of the polaron energy and number of particles in the polaron cloud is predicted for the case where the interboson repulsion is omitted or treated on the Bogoliubov level. In Ref. [74], for example, a coherent-state Ansatz for the polaron problem was introduced, yielding a polaron energy given by the simple formula

$$E = \frac{2\pi n_0}{\mu(a^{-1} - a_0^{-1})}. \tag{12}$$

Here $a_0$ is positive and arises from the interboson interactions which were included on the level of the Bogoliubov approximation. In Ref. [90] a renormalization group approach was used, predicting a divergence of the polaron energy at attractive scattering lengths. Finally, in Refs. [70, 71] we developed a Gaussian-state approach to show that large Efimov clusters can be found at energies much lower than the polaron energy, rendering the polaron a metastable local minimum in the energy landscape. At a given critical interaction strength, the polaron state loses its stability and decays into the clusters. Contrary to the coheren- state or renormalization-group approaches, the polaron energy does not "smoothly" diverge as $\frac{1}{a^{-1}-x}$, but a stable polaron just abruptly ceases to exist at a certain point, where the BEC locally collapses onto the impurity. The onset of this collapse was shown to be tied to a many-body shifted Efimov resonance, highlighting the importance of Efimov physics for this polaronic instability.

The divergence of the particle number and, in particular, the energy, should not occur in presence of realistic interboson repulsion. This immediately raises the question whether the qualitative picture of an instability can still persist in presence of realistic interboson potentials. To answer this question, we need to use a variational Ansatz allowing for an arbitrary number of particles, but also include the interboson repulsion beyond the Bogoliubov level. Coherent states including the full interboson repulsion on the level of the Born approximation have been used in Refs. [91–93]. However, a coherent state approach does not include the interboson correlations required to see the Efimov effect. Furthermore, whether this type of Ansatz is still applicable at strong interactions is questionable, since in this regime it might not capture well the atomic nature of cold gases [88]. In presence of significant repulsion between the bosons, it can be favourable for exactly one or two bosons to be around the impurity. This cannot be captured by a coherent- or Gaussian-state approach as these describe superposition states of different particle numbers, without full control over the weights of every contribution.

In this work we address these issues by comparing the Gaussian-state and double-excitation methods on equal footing, including fully the interboson repulsion beyond the Bogoliubov and the Born approximation. In this way we aim to develop a unified understanding of the Bose polaron problem and to connect the ideas and phenomenology found from all these different approaches.

# 4 Treatment of the interboson repulsion

## 4.1 Interboson repulsion energy functional

As a first step, we discuss how we describe the interboson interactions with our Gaussian-state Ansatz. Since we approximate in Eq. (2) the background BEC with a coherent state, no interboson correlations are included, and the interactions are treated on the level of the Born approximation. If we now include interboson correlations close to the impurity, the interboson interactions will be renormalized, unphysically resulting in a weaker effective interboson repulsion close to the impurity than in the background BEC.

A natural way to overcome this issue is to also treat the background BEC on the level of a Gaussian state. In this case the Gaussian part of the state would effectively perform a Bogoliubov rotation. However, this would severely complicate the structure of the cubic and quartic terms of the polaron Hamiltonian in Eq. (4).

Instead, we choose a hybrid approach. Far from the impurity we take a coherent-state wave function and describe the interactions within the Born approximation, whereas close to the impurity we keep the bare coupling, which we fully renormalize with our Gaussian-state wave function. Concretely, we achieve this by removing from the energy functional the terms responsible for renormalizing the interactions in the background BEC, which appear in the expectation values of the quadratic and cubic terms of the interboson repulsion term in Eq. (4). In the remaining quadratic and cubic terms, we replace the coupling $U$ by $U_B = \frac{8a_B}{m\sqrt{\pi}L_U}$ which gives the same scattering length on the level of the Born approximation. The quartic term is treated fully within our Gaussian-state approach. This term is the most important for the strong coupling physics close to the impurity, since it describes the repulsion between the excitations from the BEC. The precise energy functional we use is given in Appendix C.

## 4.2 Infinitely heavy impurity

To test this approach we consider first the case of an infinite-mass impurity. This is a well-studied case [88, 91–93], for which there are no impurity-mediated interactions. For negative scattering lengths up to unitarity, the Born approximation for the interboson interactions is expected to hold well [92]. As a result, a coherent state approach should describe the repulsion well when the interactions are treated using the Born approximation.

In Fig. 2 we compare our Gaussian-state result using the hybrid Born description with coherent-state results using varying interboson coupling constants (see Table 1). In the CS1 approach we fully take the Born approximation, for CS2 we take the bare coupling for the quartic term, and for CS3 we take the bare coupling in all the terms. We set $L_g = L_U = a_B$ and consider a high density BEC, $n_0 = 10^{-5}L_g^3$ (corresponding to a density of $\sim 10^{-14}$ cm$^{-3}$). We plot our results in units of the characteristic wave vector $k_n = (6\pi^2 n_0)^{1/3}$. In Fig. 2a) we plot the energies from these methods as a function of the inverse scattering length. In Fig. 2b) we plot the density of bosons as a function of the distance from the impurity. In Fig. 2c), d) and e) we show the number of excitations surrounding the impurity in a shell at a certain distance $R$:

$$\frac{\partial N}{\partial R}(R) = R^2 \int_{|\boldsymbol{r}|=R} d\Omega(\langle b_{\boldsymbol{r}}^\dagger b_{\boldsymbol{r}}\rangle - n_0), \tag{13}$$

for scattering lengths $(ak_n)^{-1} = 2$, $(ak_n)^{-1} = 0$, and $(ak_n)^{-1} = -2$, respectively.

In the weak coupling regime, i.e., in the left of Fig. 2a), the quadratic interboson repulsion term is most important, and all approaches treating this term on the same footing (GS, CS1 and CS2) agree with each other within a percent. The result from CS3 already gives a difference in energy, and furthermore, in Fig. 2e) we see that this approach underestimates the extent of

Table 1: Explanation of lines and methods used for Fig. 2. Here $U_2$ and $U_3$ stand for the coupling constants in the quadratic and cubic terms of the interboson repulsion in Hamiltonian (4) and $U_4$ stands for the quartic term. The coupling constant $U$ is the bare coupling and $U_B = \frac{8a_B}{m\sqrt{\pi}L_U}$, is the coupling giving the same scattering length on the level of the Born approximation. For more details on the modified Gaussian approach, see App. C.

| label | line | Ansatz | $U_2$ and $U_3$ | $U_4$ |
|-------|------|--------|-----------------|-------|
| GS | solid | mod. Gaussian | $U_B$ | $U$ |
| CS1 | dashed | coherent | $U_B$ | $U_B$ |
| CS2 | dash-dotted | coherent | $U_B$ | $U$ |
| CS3 | dotted | coherent | $U$ | $U$ |

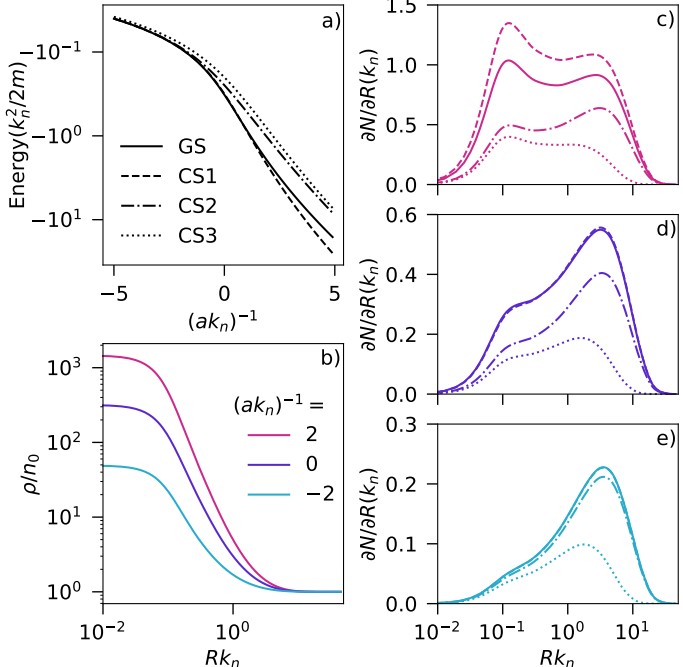

Figure 2: Properties of an infinite mass impurity immersed in an interacting BEC of density $n_0 = 10^{-5}L_g^3$ and for $a_B = L_g = L_U$. a) Polaron energy as a function of the inverse boson-impurity scattering length. b) Bosonic density for three different scattering lengths from Gaussian states as a function of the distance to the impurity. c-e) Number of additional particles at distance $R$ from the impurity [see Eq. (13)] for c) $(ak_n)^{-1} = 2$, d) $(ak_n)^{-1} = 0$, and e) $(ak_n)^{-1} = -2$. For the characterization of the computational approaches, see Tab. 1 and the main text. The legend for the line style and color in a) and b) also apply to c),d) and e).

the polaron cloud. This is because the healing length of the BEC, which sets the extent of the polaron cloud, is too small with the unrenormalized coupling constant.

Going to stronger coupling, the short-range repulsion becomes more important. While the GS and CS1 results keep agreeing with each other within 3% up to unitarity, $(ak_n)^{-1} = 0$, the CS2 approach starts to strongly deviate from these results at $(ak_n)^{-1} \approx -1$. Indeed, Fig. 2d) shows that also the differences in the wave functions become larger.

For positive scattering lengths larger than $(ak_n)^{-1} \approx 1$ the Born approximation breaks down and the Gaussian-state result starts to deviate from the CS1 result. In this regime, the

short-range repulsion is the dominant effect, since the density close to the impurity is more than a factor 1000 higher than the background density, see Fig. 2b).

## 4.3 Discussion

We now have the confidence that the Gaussian state with the hybrid Born approach properly renormalizes the interboson interactions. The Gaussian-state results namely agree with the coherent-state results with the Born approximation in the parameter regime where this approximation is valid. Without renormalization of the repulsion, the results would clearly be different, as shown via the CS2 and CS3 results. Furthermore, in the region where the Born approximation is expected to break down, indeed the Gaussian-state results are different from the coherent-state results.

Properly including the interboson interactions is more complicated for the single- and double-excitation Ansätze, since these are not mean-field approaches. Restricting the number of excitations namely implicitly leads to many-body correlations. To make the most fair comparison in the next section, we as far as possible treat the double-excitation Ansatz calculation on the same footing as the Gaussian-state calculation. The explicit form of the energy functional we use for the double-excitation approach is given in Appendix C. For the bound-state regime, where the double-excitation Ansatz is at its best, the repulsion at short range is most important, and this is described properly.

## 5 Results: Polaronic instability or smooth crossover?

We now move on to the case of a finite-mass impurity, where aside from the interboson and boson-impurity interactions, there are also *mediated* interactions between the bosons that are generated by the impurity. We will show that this new ingredient drastically changes the behaviour of the polaron. It can lead to a polaronic instability [70, 71] and induce physics akin to that of first-order phase transitions.

The reason why the polaron can become unstable is simple. If the attractive impurity-mediated interactions overcome the interboson repulsion, the net interactions between the bosons close to the impurity are attractive. Attractive BECs are known to collapse [94]. The polaronic instability is therefore nothing else than a local, impurity-induced collapse of a part of the BEC. The mechanism of the instability is described in more detail in Refs. [70, 71] as well as in Sec. 6, where we show how this physics can be qualitatively captured in a simplified analytical model.

In the present section we explore the occurrence of the instability as a function of the model parameters. We start discussing the regime of light impurities, where this instability was predicted to arise [70, 71]. Compared to these earlier studies, we now include a varying interboson repulsion and investigate its impact. Then we will study what happens as the mass ratio in the system is changed. Throughout the whole section we make a comparison of the Gaussian-state and double-excitation methods.

## 5.1 Light impurities and the polaronic instability

For concreteness, we consider a $^6$Li-impurity in a BEC of $^{133}$Cs. We vary both the impurity-boson and the boson-boson scattering lengths. We take the ranges of the potentials, $L_g$ and $L_U$, to be related via

$$\frac{L_U}{L_g} = \frac{L_{vdw,CsCs}}{L_{vdw,LiCs}}.$$ (14)

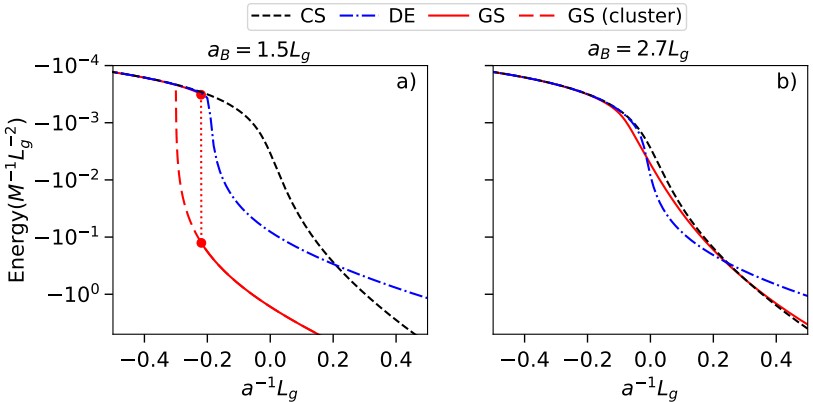

Figure 3: Polaron energies compared to cluster energies as a function of scattering length for a mass ratio $M/m = 6/133$, $n_0 = 10^{-5}L_g^{-3}$ and interboson scattering lengths of a) $a_B = 1.5L_g$ and b) $2.7L_g$, and $L_U = 2.3L_g$. The three lines correspond to the coherent-state, Gaussian-state and double-excitation Ansätze. In figure a) we observe the polaronic instability for the Gaussian Ansatz, where the energy jumps from a polaron state to a cluster state, indicated with the dots and the dotted line. The cluster state before the instability is indicated with the red dashed line, and is only shown for energies lower than the polaron energy.

Here, $L_{vdw,CsCs(LiCs)}$ is the Van der Waals length for the Cs-Cs (Li-Cs) potential. This results in a value of $L_U \approx 2.3L_g$.

First we test what our variational approaches predict for the interboson scattering lengths $a_B = 1.5L_g$ and $2.7L_g$, at a density of $10^{-5}L_g^{-3}$. In a typical scenario this density approximately corresponds to $10^{14}\text{cm}^{-3}$. We use the Gaussian-state and double-excitation approaches as discussed in the previous section. Moreover, we study a coherent-state approach using the Born approximation in all terms of the energy functional (CS1 in the nomenclature of the previous section). The results are shown in Fig. 3. Here we plot the energies of the various methods as a function of the inverse impurity-boson scattering length, in units of $L_g$. In Fig. 3a) the parameters lie in the instability regime and in Fig. 3b) in the crossover regime. One immediately sees that the curves in Fig. 3 are qualitatively different from those in Fig. 2a), and that the predictions of the three methods differ by *orders of magnitude*.

In Fig. 3a), the three approaches strongly start to differ around the Efimov resonance, since the three-body correlations introduced by the Efimov effect are treated in a widely varying manner. Before this point, on the far left of the figure, the boson-impurity coupling is weak, and all curves coincide. The coherent state does not capture three-body correlations at all, and continues describing the mean-field polaron. The double-excitation Ansatz predicts a relatively sharp crossover into the trimer state which appears at the Efimov resonance. Finally, the Gaussian-state Ansatz predicts a polaronic instability, marked in Fig. 3a) by the red dots, where the mean-field polaron ceases to be stable and decays into a many-particle cluster. The red dashed line indicates the energy of the many-body cluster before the polaron becomes unstable. In this regime the polaron is metastable [70,71] and not the ground state. Since the polaronic instability corresponds to a many-body shifted Efimov resonance [70,71], it happens close to where the trimer crosses the continuum. Compared to the Efimov energy scale, the density is still relatively low and the resonance is therefore not shifted substantially (see also Fig. 6). Since all three approaches are variational, the lowest energy state best describes the ground state, and therefore the Gaussian-state approach is most appropriate.

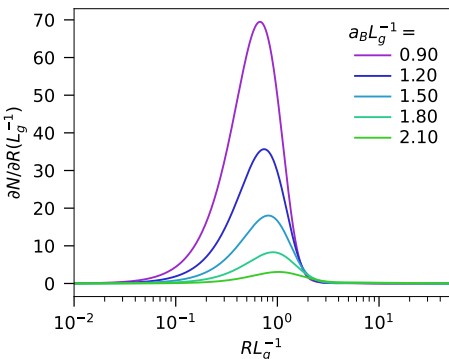

Figure 4: The wave function of the deepest bound clusters at unitarity (i.e., $a^{-1} = 0$) found from Gaussian states for the mass ratio $M/m = 6/133$ and for $n_0 = 0$. The colors of the lines indicate the value of the interboson repulsion.

In the crossover regime, such as in Fig. 3b), the story is different. Here the interboson repulsion is much larger and overcomes the mediated interactions, pushing the Efimov resonance towards unitarity. We therefore see that the mean-field regime, where the curves coincide, extends to stronger interactions. Around unitarity, now the double-excitation Ansatz gives the best description. In this regime, no large clusters can be formed, and the polaron experiences a crossover into a trimer-like state. As evident, this behaviour cannot be captured well by the Gaussian- or coherent-state methods, whose curves lie relatively close together. Only when the attractive impurity-boson interaction is increased further, more particles can bind to the impurity, and the Gaussian- and coherent-state methods outperform the double-excitation approach, as signified by their resulting energies. Note, however, that this occurs relatively far outside the universal regime.

The character of the clusters that form in the instability regime in Fig. 3a) is quite distinct from the polaron. Example wave functions of the clusters are shown in Fig. 4 for increasing interboson repulsion in absence of a background BEC ($n_0 = 0$) and for unitary boson-impurity interactions. We see in Fig. 4 that many particles come together within the range of the potential. A polaron cloud can also host many particles, but for a polaron state most particles are far away from the impurity, at a distance set by the healing length (see Fig. 2c-e)).

For increasing repulsion, the number of particles in the cluster rapidly decreases. At some point the bound state contains only a few particles. Here the Gaussian-state description (as the coherent-state approach) fails since it is bound to represent a superposition of states with different particle numbers, with limited control over the weights of their contributions. This is especially detrimental for the description of a bound state with exactly one or two particles. Therefore, a double-excitation Ansatz is more suitable in this regime.

As a technical side note, numerically, we find not just one, but in fact two types of stable cluster states from the Gaussian-state approach. The clusters shown so far contain both a coherent and a Gaussian contribution to the wave function, but another local minimum on the variational landscape arises when there is solely a Gaussian contribution to the wave function ($\phi = 0$). This second type of clusters is generally higher in energy, but the real-space wave functions such as in Fig. 4 are qualitatively similar. There is one regime where the second type of cluster is lower in energy than the first type: when the particle number in the cluster goes to zero. In this case the Gaussian state just describes a superposition of the free impurity and a trimer.

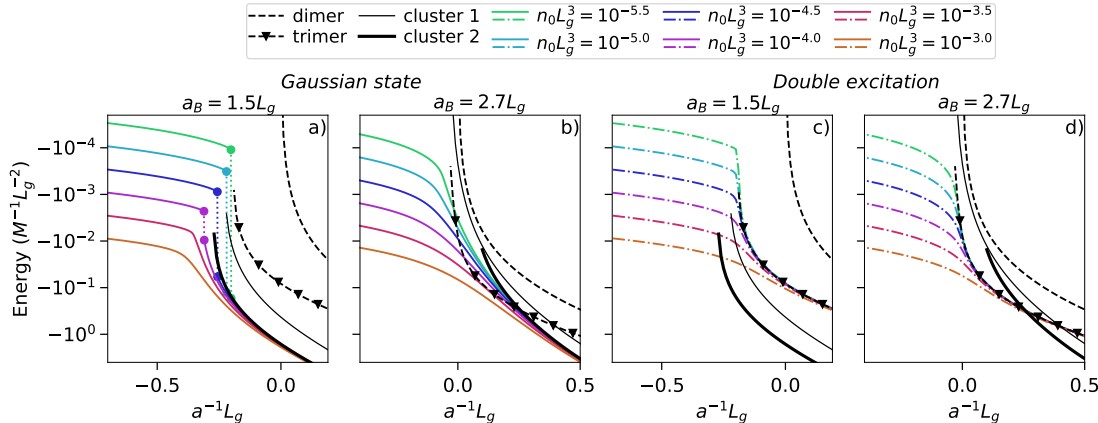

Figure 5: Polaron energies (colored lines) as a function of the inverse boson-impurity scattering length for a light impurity ($M/m = 6/133$), calculated using a,b) the Gaussian-state Ansatz, c,d) the double-excitation Ansatz. In panels a) and c) $a_B = 1.5L_g$, in panels b) and d) $a_B = 2.7L_g$. The color of the lines indicates their corresponding background density. In all panels the black dashed lines indicate the dimer and trimer (with triangles) energies. The black solid lines indicate the energies from the two types of cluster from the Gaussian-state approach at $n_0 = 0$. In figure a) the point of polaronic instability is indicated with the pairs of dots and dotted lines.

## 5.2 Emergent "phase diagram"

We now discuss the behaviour of the polaron as predicted from the variational methods, for varying densities of the background BEC. In Fig. 5 we show the energy of the polaron as a function of the inverse scattering length for several densities (indicated by the colors). We have also added the energies of the dimer (black dashed), trimer (black dashed with triangles) and the two types of cluster discussed before (thin and thick black solid) as a function of the inverse scattering length in absence of a background BEC. The different panels correspond to Gaussian-state results [Fig. 5a) and b)] and double-excitation results [Fig. 5c) and d)] for different values of the interboson repulsion. Note that the scale of the x-axis is different in the left and right panels.

For weak impurity-boson attraction, on the left side of all four panels, the polaron is in the mean-field regime, and the energy should depend approximately linearly on the density [see Eq. (12)]. Since the colored lines are spaced linearly on the logarithmic energy grid and the corresponding densities are also spaced logarithmically, we indeed recover this expected result. The energy scale of the polaron is much smaller than the scale of the bound states, except for the highest densities.

In Fig. 5a) we again note the presence of the polaronic instability, but interestingly, this instability disappears for large densities. The transition from a polaronic instability to a crossover can therefore happen both as a function of density, and as a function of the interboson repulsion. The point where the instability disappears is the point where the gap between the polaron energy and the cluster energy closes. In the case of Fig. 5a), this gap is closed by increasing the polaron energy through an increase in the density. The character of the cluster is largely unaffected by the increase of the density, meaning that the formed clusters can still contain many particles even in the crossover regime.

In Fig. 5b), the repulsion is large enough that there is a crossover for all densities. For all but the largest densities, the Gaussian state does not describe this crossover well around unitary interactions [compare to Fig. 5d)] since the Gaussian-state energy actually lies above

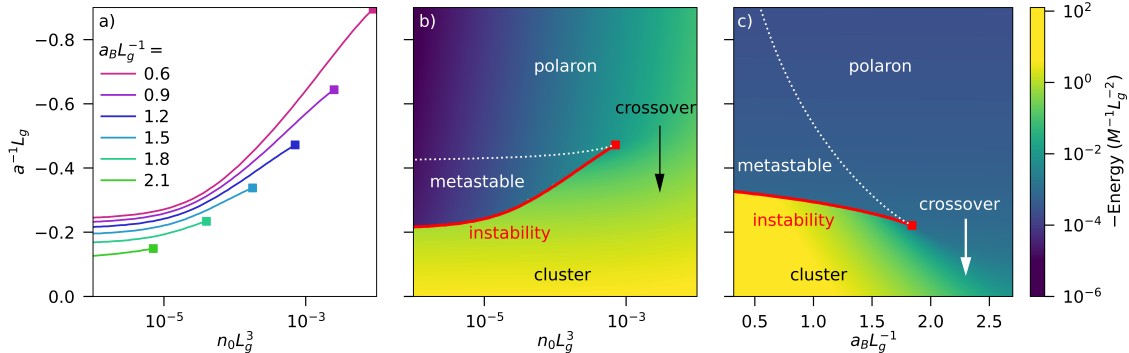

Figure 6: a) Critical scattering length $a_c$ of the polaronic instability as a function of the density $n_0$, for various values of the interboson scattering length $a_B$. At the square endpoints of the line the gap between the polaron and cluster states has closed, and the instability turns into a crossover. b,c) Polaron energy as a function of scattering length for b) varying density and fixed $a_B = 1.2L_g$ and c) varing $a_B$ and fixed density $n_0L_g^3 = 10^{-4.5}$. The red lines indicate the critical scattering length of instability, such as in figure a). The white dotted line indicates the scattering length where the energy of the cluster crosses the polaron energy. Between the dotted and dashed lines the polaron is thus metastable. The mass ratio is chosen to be $M/m = 6/133$ and $L_U = 2.3L_g$.

the trimer energy.

In the lower two panels [Fig. 5c) and d)], we see that the double-excitation Ansatz behaves qualitatively the same for all densities and for both values of the interboson repulsion: it gives a smooth crossover between the polaron and the trimer state, irrespective of the presence of larger clusters, which cannot be captured by this approach. It is interesting to notice that for increasing density, the crossover happens over a broader range of interaction strengths, and that the interaction strength where the polaron energy merges with the trimer line becomes larger. This is the opposite trend compared to the instability in Fig. 5a), where the polaronic instability happens for decreasing interaction strengths as the background density is increased. This difference can be explained as follows. For the Gaussian state, a larger particle number in the polaron cloud means that it is easier to decay into a cluster. Hence, at larger densities smaller interaction strengths are needed. For the double-excitation Ansatz, there is a competition between the polaron energy, which increases with the density, and the trimer energy, which is not strongly affected by the density. Therefore, the regime of the crossover is shifted towards larger interaction strengths as the density increases.

Next, we examine in more detail the density dependence of the critical scattering length of the polaronic instability. In Fig. 5a) we can already see that the point of instability shifts to smaller scattering length as the density increases. In Fig. 6a), this critical scattering length is plotted as a function of the density for various values of the interboson repulsion, increasing from top to bottom.

For low densities, on the left hand side of Fig. 6a), the lines of critical scattering length approach the value of the Efimov scattering length [70, 71]. This scattering length already gives a dependence on the interboson repulsion, reflecting that also the value of the Efimov scattering length depends strongly on the interboson repulsion.

When moving to higher density, the critical interaction strength (at fixed repulsion) decreases and at some critical density the gap between the polaron and cluster energies closes. This is also visualised in Fig. 6b), where the underlying energy landscape (computed using Gaussian states) is shown as a colormap for $a_B = 1.2L_g$. Here the instability is marked with

the red line. The point where the gap between polaron and cluster closes, marks the end point of the transition from the polaron to the cluster states. Beyond this density a crossover occurs. Looking again at Fig. 6a), we find that the density where the instability terminates increases rapidly with the interboson repulsion. This is because the cluster binding energy decreases with the repulsion. Therefore the gap closes already at smaller polaron energies and thus smaller densities.

Aside from the red line marking the instability, Fig. 6b) also shows (white dotted line) the scattering length where the cluster energy first crosses the polaron energy. This line therefore marks the point where the ground state of the model truly switches character. In between the white dotted and red lines we find the regime of metastability, where the polaron is not the ground state, but where there is no decay process included in our Ansatz from the polaron state into the cluster state. In Fig 3 this is the region before the instability where the dashed line lies below the solid line. Note that the white dotted line is almost horizontal in this figure. That is because the cluster energies are on a different scale than the polaron energy, and changing the polaron energy via the density therefore does not strongly affect the crossing point of the polaron and cluster energies.

In Fig. 6c) we show a similar graph, but here we fix the density $n_0 L_g^3 = 10^{-4.5}$ and vary instead the interboson repulsion. While as a function of the density, the polaron energy changes and the cluster energy is approximately constant, the opposite is true when varying the interboson repulsion, which has a much larger impact on the cluster state than the polaron. Therefore in this case the gap between the polaron and cluster energies is closed by decreasing the cluster energy. However, qualitatively the picture is the same. There is still a transition terminating in a critical point, beyond which there is a crossover. As we saw in Fig. 6a), the point of instability shifts to larger interaction strengths for increasing repulsion, following the trend of the position of the Efimov resonance.

In contrast to Fig. 6b), where the white dotted line is almost flat, here the white dotted line varies much more with the interboson repulsion than the line of instability. This is because the line of instability is determined by the smallest cluster into which the polaron can initially decay, and the line of metastability by the many-particle cluster which is lowest in energy. Since the more particles there are in the cluster, the more important their repulsion, it is not surprising that the metastability line depends more strongly on the repulsion than the instability line.

Altogether, we see that a remarkable picture emerges of a first-order transition between a polaron and a cluster, which terminates at a critical point. This is strongly reminiscent of classical first-order phase transitions, such as the phase transition of condensation of a gas into a liquid. We discuss this analogy in more detail in Sec. 6 where we develop an analytical model for the Bose polaron showing the same qualitative behaviour.

## 5.3 Mass dependence

So far we have considered the scenario of a light impurity in a BEC, where the impurity-mediated interactions are particularly strong. Now we explore in more detail how the phenomena we observe manifest themselves for a wider range of impurity masses.

In Fig. 7, diagrams are shown that indicate for which values of the mass ratio and interboson repulsion the polaronic instability appears. The color code gives the corresponding critical scattering length. The background density $n_0$ is fixed within both panels, and given by a) $10^{-6} L_g^{-3}$ and b) $10^{-4} L_g^{-3}$ (approximately $10^{13}$ and $10^{15}$ cm$^{-3}$ respectively). In Fig. 1b) another such colormap is shown for the density $10^{-5} L_g^{-3}$. Note that on the x-axis the interboson scattering length is given in units of $L_U$. We have chosen these units, because varying $L_U$ while keeping $a_B/L_U$ fixed only gives rise to minor changes in the plots. This indicates that

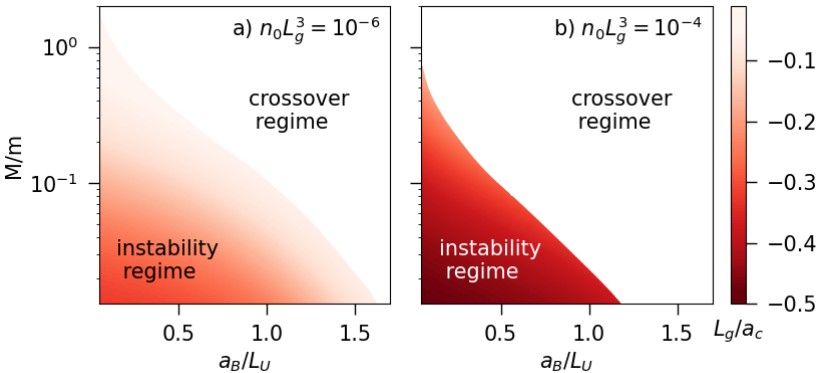

Figure 7: Stability diagram of the polaron as a function of the impurity-boson mass ratio $M/m$ and the interboson repulsion scattering length $a_B$. In the lower left part of the diagram, when the boson impurity scattering length is swept across the Feshbach resonance, the polaron will experience an instability at scattering length $a_c$ indicated by the colormap. In the upper right part the polaron will smoothly cross over into a small cluster. For this plot $L_g = L_U = 1$ and $n_0 L_g^3 = 10^{-5}$.

also the range of the interboson repulsion matters, and not just its scattering length. This is not surprising, since the range of the potentials is known to be important for the Efimov effect.

We see that the region of instability appears for light impurities and weak interboson repulsion, in the lower left of both panels. Here, the impurity-mediated interactions are the strongest and dominate over the interboson repulsion. For equal mass or heavy impurities the regime where an instability appears shrinks drastically. Furthermore, no instability appears if $a_B$ is significantly larger than $L_U$.

Comparing Figs. 7a) and b), we observe that the regime of instability shrinks as the density is increased. This is because the larger the density, the larger the polaron energy, and hence the smaller the energy gap between polaron and cluster. From the change of color one sees that the critical scattering length at which the instability occurs also becomes smaller (as also visible in Figs. 5a) and 6).

## 5.4 Comparison of Gaussian-state and double-excitation Ansatz

Having discussed the qualitative distinction between the regimes of the instability and the crossover, we now systematically compare the Gaussian-state and double-excitation methods. The main aim of this comparison is to demonstrate in which parameter regime which method is best to use. In Fig. 8 we show the polaron energies from the Gaussian-state Ansatz (first column), the double-excitation Ansatz (second column), and their ratio, as a function of the mass ratio and the interboson repulsion. The different rows correspond to different boson-impurity scattering lengths. Comparing to Fig. 7, the $y$-axis extends to larger mass ratios.

In the lower left of the plots, again the region of polaronic instability appears. Here the energy found from Gaussian states is obviously lower than the energy from the double-excitation Ansatz, as evident from the right column and the results we have shown before. The region in parameter space where a many-particle cluster is found with Gaussian states is more or less similar to the regime where a trimer is formed with much lower energy than the energy scale of the mean-field polaron, as seen from the double-excitation results.

For system parameters in the crossover regime, the Gaussian-state and double-excitation results appear more similar, because here at least the energy scale is the same: the scale of the mean-field polaron energy, set by the density of the BEC. For heavy impurities and

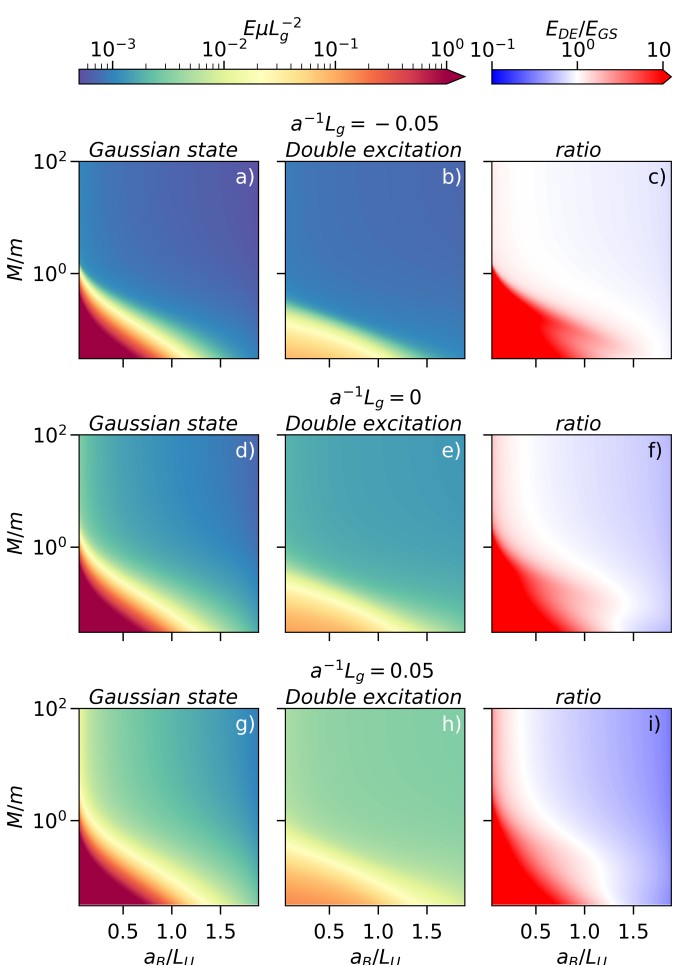

Figure 8: Colormaps of the polaron energy as a function of the mass ratio $M/m$ and interboson scattering length $a_B$ from the Gaussian-state Ansatz (first column) and double-excitation Ansatz (second column). In the third column the ratio of the energies from these approaches is shown. The different rows correspond to different inverse boson-impurity scattering lengths, given by a-c) $-0.05L_g^{-1}$, d-f) $0L_g^{-1}$, f-i) $0.05L_g^{-1}$. Here $L_g = L_U$ and the density is given by $n_0 = 10^{-5}L_g^{-3}$.

$a = -0.05L_g^{-1}$, in the blue region of Fig. 8a) and b), the system is truly in the polaron regime, and bound state physics does not play a crucial role. Here the ground state energy (which is already rescaled by the reduced mass) is only mildly dependent on the mass ratio and the interboson repulsion. Despite being in the polaron regime, in the far left and far right of the plot there is a significant energy difference of up to 30 % between the Gaussian-state and double-excitation results. For low interboson repulsion the Gaussian-state performs better, because here many excitations can come close to the impurity. In contrast, in the regime of large repulsion the double-excitation Ansatz works better. Here the number of excitations is limited, and having more correlations between these few excitations and the bath is therefore more effective. Note that this argument even holds for the infinitely heavy impurity. Thus, our results show that merely the Born approximation being satisfied is not sufficient to show that a coherent or Gaussian state accurately describes the ground state. This has the important implication that also the Gross-Pitaevskii equation, which is equivalent to a coherent state approach, loses its validity. This confirms the analysis of Ref. [88].

When moving from the upper row of Fig. 8 to the lower rows, i.e., to stronger boson-

impurity interaction strengths, we see that the differences between the Gaussian-state and double-excitation results become progressively larger. This is because here bound-state physics starts to play a larger role. In the right-hand side of the plots a bound state containing only one or two particles is more favorable than a true polaron state. This is reflected in the much lower energy of the double-excitation result compared to the Gaussian-state result. However, towards the left hand side of these plots the number of particles close to the impurity grows significantly, rendering, in turn, the double-excitation Ansatz insufficient.

In the strong-coupling regime where Gaussian states and the double-excitation Ansatz give similar energies [white regions in figures f) and i)], we expect that actually neither of them work very well. While the double-excitation Ansatz does not have enough excitations, the Gaussian-state Ansatz does not give enough independent control over the different particle number sectors. We identify this as the most challenging regime, which requires going beyond the double-excitation Ansatz [69] or using Quantum Monte Carlo.

## 5.5 Experimental implementation

Now we turn to a discussion of the feasibility of realizing the physics predicted in this work in experiments. First, let us discuss the density range of the BEC. The density range of $10^{-6}$-$10^{-4}L_g^{-3}$, corresponds via $L_g \approx 2L_{vdw}$ and a typical value of $L_{vdw} \approx 50a_0$ [8] to densities of $7 \cdot 10^{12}$-$10^{14}$ cm$^{-3}$. Such densities are readily available experimentally.

Next, we turn to the interboson repulsion. With the usual magnetic-field controlled Feshbach resonances, the boson-impurity and boson-boson scattering lengths can not be tuned independently. This limits experiments to the background scattering lengths in the BEC at the positions of the boson-impurity resonances. Far away from Feshbach resonances and for a stable BEC, typically $a_B \sim L_{vdw}$. As a result, most experiments without large mass imbalance naturally operate relatively deep in the crossover regime (see Figs. 1 and 7).

There are other mechanisms to change the scattering lengths, which could be used in combination with the magnetic field approach to be able to tune multiple scattering lengths simultaneously [95]. Optical control of the scattering length has already been observed [96, 97], and Feshbach resonances tuned via radiofrequency or microwave fields [95,98] have been proposed theoretically. Even though these schemes usually lead to losses, in the Bose polaron context it might help that the interboson scattering length needs to be decreased instead of increased, and that experiments would only need to run for a short time.

From the Bose polaron experiments carried out so far, the experiments using $^{39}$K [59,62] incidentally have a very small interboson scattering length of $a_B \approx 9a_0$. While this is most likely not in the regime of instability, it is relatively close, and therefore the formation of large clusters could be expected. However, to populate such deeply bound states one needs to go beyond the standard injection spectroscopy, because starting with a non-interacting state will give only very small overlap with such deep, many-particle bound states.

Concerning the realization of varying mass ratios between the impurity and the bosons, there are setups realizing a large mass imbalance with light impurities. Prominent examples are the Li-Cs mixtures used to study heteronuclear Efimov physics [99–102]. Unfortunately, for $^{133}$Cs the background scattering length is usually very large, so this would most likely prevent the presence of the polaronic instability.

Another issue which arises in systems of large mass imbalance is very fast three-body loss at strong coupling. This loss arises precisely from the same mediated interactions between bosons which also lead to the polaronic instability. While this highlights another interesting connection between the field of polarons and conventional few-body physics, this may also make preparing such mixtures in a stable way at ultracold temperatures difficult.

Altogether, the physics which would be observed could differ tremendously between the

different mixtures and even for different Feshbach resonances in the same mixture, see below. This gives the exciting opportunity to explore the different types of behaviour. However, this also implies that caution is required when comparing different experiments, since seemingly small differences in the experimental parameters might lead to drastically different behaviour. This gives a unique opportunity to explore the rich interplay between few-and many-body physics.

## 5.6 Discussion

We now discuss some limitations of the theoretical approaches employed in this work.

**Description of the clusters**

A Gaussian state is not very suitable to accurately describe the structure and energies of the deeply bound clusters formed in the instability regime. There are several reasons for that. First, in absence of a background condensate, the true cluster states are particle number eigenstates and Gaussian states are not. In fact, the spread in the particle number of a Gaussian state is of the order of the number of particles. Second, higher-order correlations will most likely become important in the microscopic description of many-body clusters. Even though three-body correlations between the bosons and the impurity are included, this will generally not be sufficient for bound states of more than three particles. Third, the properties of deep many-particle bound states can not be expected to be universal. Therefore the properties of these states will depend on the details of the interactions potentials and the use of simple model potentials is not warranted.

However, the structures of these clusters are unlikely to be important for experimental observables. Experimentally, once such a deeply bound cluster is formed, this would immediately lead to fast recombination losses. Furthermore, in the density regime reachable in cold-atom experiments, the qualitative mechanism of the polaronic instability should not be very sensitive to the detailed structure of the underlying clusters. In fact, unlike the wave function of the deeply bound clusters, the mechanism of the instability itself and the mediated interactions should be universal. The mediated interactions namely originate from the Lee-Low-Pines term in the Hamiltonian, which is independent of any potential. Furthermore, the first clusters into which the polaron decays will contain only few particles, and only after the formation of these small intermediate clusters, will the system cascade into the deeply bound ground state. When the background density of the BEC becomes very high this picture might break down, however, and here the point where the instability terminates will depend on the actual energy of the cluster.

**The double-excitation Ansatz**

In the crossover regime of Figs. 1 and 7, far enough away from the polaronic instability, we believe that the ground state energy is well described by the double-excitation Ansatz. Possible corrections can be accounted for by an extension to the triple-excitation Ansatz [69]. Therefore, we believe that regarding the ground-state energies, the picture sketched for equal mass and heavy impurities is accurate in Refs. [69, 88].

However, whether the *wave function* of the resulting state is also well described, remains an open point of discussion [74]. For example, in the regime where the ground state of the system is a trimer, the double-excitation Ansatz will 'invest' its excitations to describe this trimer state. However, it is not unlikely that in reality this molecule will again induce its own polaron cloud. The double-excitation Ansatz is not capable of describing this behaviour, since its excitations are already used up in the description of the bound state itself. While for the ground-state

energy this may not have a significant effect, for the wave function and the description of the full excitation spectrum it can certainly be of importance.

Another short-coming of the double-excitation Ansatz is its description of the weak-coupling Bose polaron in presence of interboson repulsion. For the weak coupling case, it is most natural to describe the polaron cloud on the same footing as the background BEC. This is naturally achieved using a coherent- or Gaussian-state Ansatz. In these approaches it is also more clear how to describe the interboson interactions correctly. Consistently doing this with the double-excitation Ansatz is more difficult, because this would require an accurate description of the interactions in the background BEC.

Moreover, the double-excitation Ansatz brings with it the subtle problem of how to count the number of particles in the polaron cloud. One would expect that the double-excitation Ansatz gives rise to at most two excitations. This is indeed true if one does not allow the double-excitation Ansatz to have a component in the mode of the BEC. However, with a coherent-state or Gross-Pitaevskii approach, one does not have this restriction. If indeed the double-excitation Ansatz is also allowed a component in the mode of the BEC, then the cross-terms with the BEC will give rise to equally large particle numbers in the polaron cloud as in the coherent-state case. This could potentially explain the qualitative difference between the large number of particles the coherent-state approaches predict in the polaron cloud [91–93] and the apparent success of variational approaches with only few excitations [67, 69, 88].

**Narrow Feshbach resonances**

In this work we use a single-channel model for the boson-impurity interactions, which is best suited to describe broad and isolated Feshbach resonances [8]. In Refs. [67, 69] a two-channel model has been used, which is also applicable to narrow resonances. In this case, the multi-channel nature of the interactions can lead to an effective three-body repulsion. This will have a similar effect as the intrinsic interboson repulsion and therefore help to suppress the instability. This may therefore lead to a shift of the boundary of the instability region in Figs. 1 and 7.

# 6 Analytical model

The form of Fig. 6b) and c), with its first-order transition ending in a critical point, followed by a smooth crossover, is remarkably similar to well-known diagrams of first-order phase transitions such as the liquid-gas phase transition. To strengthen this connection we attempt to understand the Bose polaron phenomenology in simpler terms. To this end, we develop a analytical model to qualitatively reproduce the key features of Fig. 1 and Fig. 6. Surprisingly, we find that we can achieve this with a much simplified Gaussian-state Ansatz. As we will see, even when restricting the variational Ansatz to only $l = 0$ and $l = 1$ angular momentum modes, and only a *single* Gaussian basis function per angular momentum mode, the model already qualitatively reproduces the important physics.

We thus start the derivation of the analytical model by writing

$$\phi(\mathbf{k}) = \phi \chi_{00}(\sigma_\phi, \mathbf{k}), \tag{15}$$

$$\xi(\mathbf{k}, \mathbf{q}) = \xi_0 \chi_{00}(\sigma_\phi, \mathbf{k}) \chi_{00}(\sigma_\phi, \mathbf{q}) + \xi_1 \sum_m (-1)^m \chi_{1m}(\sigma_1, \mathbf{k}) \chi_{1-m}(\sigma_1, \mathbf{q}). \tag{16}$$

## 6.1 Coherent states and effective scattering length

To build our understanding, we start by first studying a coherent-state Ansatz and omitting the interboson interactions in the Hamiltonian. We replace the real-space Gaussian boson-impurity potential by a short-range separable interaction with a Gaussian cutoff function in momentum space,

$$\hat{\mathcal{H}}_{\text{int}} = g\Big[\int_{\boldsymbol{k}} e^{-\sigma_g k^2}(\hat{b}_{\boldsymbol{k}}^{\dagger} + \sqrt{n_0}\delta(\boldsymbol{k}))\Big]\Big[\int_{\boldsymbol{k}} e^{-\sigma_g k^2}(\hat{b}_{\boldsymbol{k}} + \sqrt{n_0}\delta(\boldsymbol{k}))\Big]. \tag{17}$$

The scattering length $a$ for this potential is related to the coupling constant $g$ as

$$g^{-1} = \frac{\mu_r}{2\pi a} - \frac{\mu_r}{\sqrt{(2\pi)^3 \sigma_g}}. \tag{18}$$

Using only the coherent state part of our Ansatz (i.e. $\xi = 0$), one finds the energy as a function of $\phi$ and $\sigma_\phi$ to be given by

$$E(\phi, \sigma_\phi) = gn_0 + T_\phi \phi^2 + 2gV_\phi \sqrt{n_0}\phi + gV_\phi^2 \phi^2, \tag{19}$$

where $T_\phi$ and $V_\phi$ are the expectation values of the kinetic and interaction energies

$$T_\phi = \int_{\boldsymbol{k}} \frac{k^2}{2\mu_r} |\chi_{00}(\sigma_\phi, \boldsymbol{k})|^2 = \frac{3\sqrt{\pi}}{16\mu_r(2\sigma_\phi)^{5/2}}, \tag{20}$$

$$V_\phi = \int_{\boldsymbol{k}} e^{-\sigma_g k^2} \chi_{00}(\sigma_\phi, \boldsymbol{k}) = \frac{1}{4\sqrt{2\pi(\sigma_g + \sigma_\phi)^3}}. \tag{21}$$

Eq. (19) can be trivially minimized with respect to $\phi$ and $\sigma_\phi$. The resulting value of $\sigma_\phi = 5\sigma_g$ is independent of any of the other parameters. This leads to

$$E = \frac{n_0}{g^{-1} + \frac{V_\phi^2}{T_\phi}} = \frac{2\pi n_0}{\mu_r(a^{-1} - a_{\text{shift}}^{-1})}, \tag{22}$$

where

$$a_{\text{shift}}^{-1} = (2\pi\sigma_g)^{-1/2} - \frac{2\pi V_\phi^2}{\mu_r T_\phi} = \frac{1 - \frac{5^{5/2}}{3^4}}{\sqrt{2\pi\sigma_g}}. \tag{23}$$

Note that this expression of the energy is remarkably similar to the energy found from mean field theory assuming a weakly repulsive BEC within the Bogoliubov approximation [74]. In the case of Eq. (12) from Ref. [74], the origin of the shift $a_0$ of the scattering length in the denominator can be traced to the interboson repulsion limiting the size of the polaron cloud. In our case it is the exponent of the Gaussian basis function that limits the size of the cloud.

If we want to compare our analytical result with the full model we can define an effective scattering length

$$a_{\text{eff}}^{-1} = a^{-1} - a_{\text{shift}}^{-1}. \tag{24}$$

This effective scattering length diverges when a bound state appears in our model. We can also write $g$ in terms of $a_{\text{eff}}$ as

$$g^{-1} = \frac{\mu_r}{2\pi a_{\text{eff}}} - \frac{V_\phi^2}{T_\phi}. \tag{25}$$

With this replacement, the coherent state energy for the polaron without background repulsion is recovered perfectly.

## 6.2   Gaussian states and the polaronic instability

Having considered the coherent state case, we now include also the Gaussian part as in Eq. (16) into the wave function, still without including the interboson repulsion. The additional variational parameters are $\xi_0$, $\xi_1$, $\sigma_0$ and $\sigma_1$. We keep $\sigma_\phi = 5\sigma_g$ fixed, since the coherent part is the dominant part in the polaron regime. Expanding the energy functional up to quadratic order in $\phi^2$, $\xi_0$ and $\xi_1$, we find

$$E(\phi, \xi_0, \sigma_0, \xi_1, \sigma_1) = g n_0 + T_\phi \phi^2 + (T_0 + g V_0^2) S_0 \xi_0^2 + 3 T_1 S_1 \xi_1^2 + 2 g V_\phi \sqrt{n_0} \phi + g V_\phi^2 \phi^2$$
$$- 3 T_{L\phi} \phi^2 \xi_1 - 3 T_{L0} \xi_0 \xi_1. \quad (26)$$

In this expression, the quantities $T_1$ and $T_{L\phi}$ are given by

$$T_1 = \int_k \frac{k^2}{2\mu_r} |\chi_{1m}(\sigma_\phi, k)|^2 = \frac{15\sqrt{\pi}}{32\mu_r (2\sigma_1)^{7/2}} \quad (27)$$

$$T_{L\phi} = -\frac{1}{M} \left[ \int_k k \chi_{1m}^*(\sigma_1, k) \chi_{00}(\sigma_\phi, k) \right]^2 = \frac{3\pi}{64 M (\sigma_\phi + \sigma_1)^5}. \quad (28)$$

The quantities $T_0$, $V_0$ and $T_{L0}$ are defined similarly to $T_\phi$, $V_\phi$, and $T_{L\phi}$ by substituting $\sigma_\phi$ by $\sigma_0$. The $T_{L\phi}$ and $T_{L0}$ terms originate from the Lee-Low-Pines term in the Hamiltonian.

The terms $S_0$ and $S_1$ are overlap integrals, given by

$$S_0 = \int_k |\chi_{00}(\sigma_0, k)|^2 = \frac{\sqrt{\pi}}{4(2\sigma_0)^{3/2}}, \quad (29)$$

$$S_1 = \int_k |\chi_{1m}(\sigma_0, k)|^2 = \frac{3\sqrt{\pi}}{8(2\sigma_1)^{5/2}}. \quad (30)$$

Minimizing the energy functional (26) with respect to $\sigma_0$, $\sigma_1$, $\xi_0$ and $\xi_1$ now yields $\sigma_0 = \sigma_\phi = 5\sigma_g$ and $\sigma_1 = \frac{15}{2}\sigma_g$. This implies $T_\phi = T_0$ and $T_{L\phi} = T_{L0} = T_L$. For $\xi_0$ and $\xi_1$ one then finds

$$\xi_0 = \frac{3 T_L}{2 S_0 (T_0 + g V_0^2)} \xi_1, \quad (31)$$

$$\xi_1 = \frac{T_L \phi^2}{2 S_1 T_1 - \frac{3 T_L^2}{2 S_0 (T_0 + g V_0^2)}}. \quad (32)$$

The parameters $\xi_i$ characterize the Gaussian part of the wave function. Thus, already from Eqs. (31) and (32) we can see a sign of the instability, which will occur when the denominator of $\xi_1$ vanishes and hence $\xi_0$ and $\xi_1$ diverge. From Refs. [70, 71] we know that in the low-density limit this happens at the Efimov scattering length $a_-$, where the three-body Efimov bound state arises. Hence, if we derive the value of $a_{\text{eff}}$ where the divergence occurs, we can extract the value of $a_{\text{eff},-}$ in our model with the simplified wave function. This is given by:

$$a_{\text{eff},-} = \frac{T_0 \mu}{V_0^2 2\pi} \left(1 - \frac{4 S_1 T_1 S_0 T_0}{3 T_L^2}\right) \quad (33)$$

$$= \frac{3^4 \sqrt{2\pi\sigma_g}}{5^{5/2}} \left(1 - \frac{M^2 5^{11}}{\mu_r^2 2^{14} 3^6}\right) \approx 3.6 \sqrt{\sigma_g} \left(1 - 4.1 \frac{M^2}{\mu_r^2}\right) \quad (34)$$

The scattering length $a_{\text{eff},-}$ is negative. We thus see that even in our simple model a three-body bound state appears from the continuum in a Borromean way, i.e., it arises before a

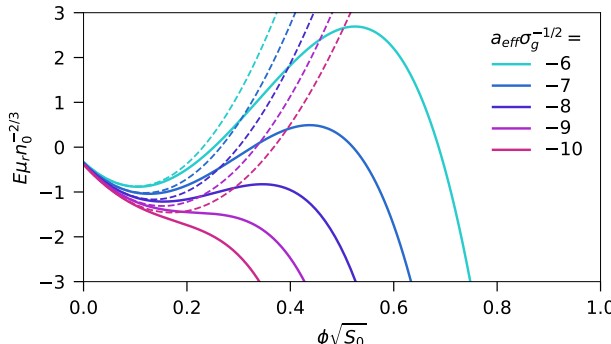

Figure 9: Energy functional of Eq. (22) (dashed) and Eq. (35) (solid), where the energy is plotted in units of $\mu_r n_0^{-2/3}$ as a function of the normalized "order parameter" $\phi$ for several effective scattering lengths. The magnitude of the scattering length increases from top to bottom. The mass ratio here corresponds to Li-Cs and $n_0 \sigma_g^{3/2} = 1.25 \, 10^{-5}$.

two-body bound state is possible. Furthermore, the linear scaling of $a_{\text{eff},-}$ with the length scale of the potential $\sqrt{\sigma_g}$, behaves according to the expectations of van der Waals universality [82–84]. Finally, note that for a light impurity $\frac{M^2}{\mu_r^2} \to 1$ and for a heavy impurity $\frac{M^2}{\mu_r^2} \to \infty$. This implies that $a_{\text{eff},-} \to -11.2\sqrt{\sigma_g}$ for light impurities and $a_{\text{eff},-} \to -\infty$ for heavy impurities. The limits of the mass-dependence of $a_{\text{eff},-}$ are therefore also physical. Quantitatively, however, one should note that the realistic mass dependence of $a_-$ is stronger than found here.

Now we can plug in the results for $\xi_0$ and $\xi_1$ to obtain an energy functional in terms of only $\phi$:

$$E(\phi) = g n_0 + 2g V_0 \sqrt{n_0}\phi + (T_0 + g V_0^2)\phi^2 - \frac{\mu_r S_0 T_0^2 \phi^4}{2\pi V_0^2(a_{\text{eff}} - a_{\text{eff},-})}. \tag{35}$$

The structure of this equation with a linear, quadratic and quartic term in $\phi$ is reminiscent of the paradigmatic Landau model for phase transitions, where $\phi$ would correspond to the order parameter. Our scenario, with a positive quadratic term and a negative quartic term, corresponds to the case of a *first-order* phase transition. Here the linear term, which depends on the density of the background BEC, adds an effective external field to the description (similar to a external magnetic field in the theory of phase transitions in magnetic materials). Note that all the terms depend explicitly on the boson-impurity interaction strength.

The energy functional (35) is plotted in Fig. 9 for different values of $a_{\text{eff}}$. Here $\sqrt{S_0}$, as defined in Eq. (29), serves to normalize the Gaussian basis function. The dashed lines indicate the combined contribution of the linear and quadratic parts, while the solid lines show the full result from Eq. (35).

For small $a_{\text{eff}}$ the function has a minimum at small $\phi$ corresponding to the polaron state. In this regime, the quartic term plays no role yet. For increasing $a_{\text{eff}}$ the value of $\phi$ at the polaron minimum increases. Therefore the quartic term becomes more and more important. As evident from Eq. (35), the quartic term is also directly dependent on $a_{\text{eff}}$, further enhancing its importance for increasing $a_{\text{eff}}$.

At some point the quartic term overcomes the quadratic term and the polaron minimum disappears: the polaron becomes unstable. Because no interboson repulsion is included to stabilize the energy functional, $\phi$ will grow indefinitely beyond this point. Our model breaks down in this limit since $\xi_0$ and $\xi_1$ cease to be small parameters, and higher order terms in $\xi$ will be important in Eq. (26).

The point of polaronic instability does *not* correspond to the point of a phase transition in

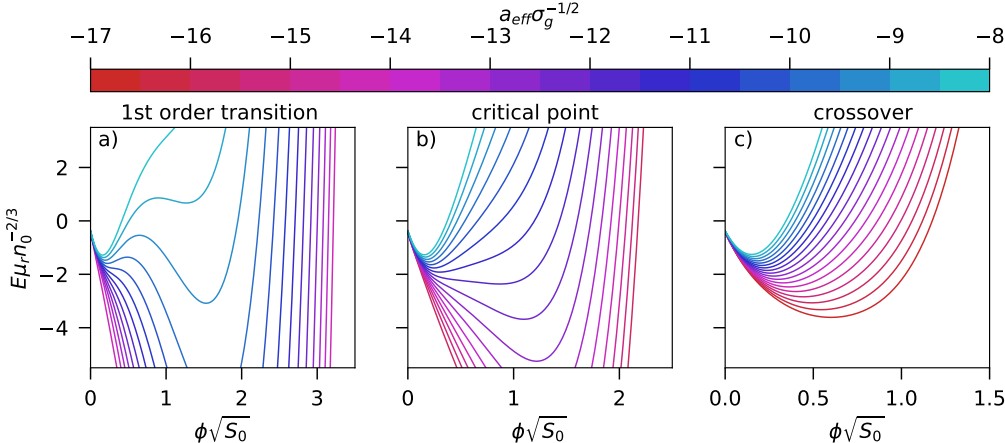

Figure 10: Energy functional of Eq. (44), where the energy is plotted in units of $\mu_r n_0^{-2/3}$ as a function of the normalized $\phi$ for several scattering lengths. The magnitude of the scattering length is indicated by the colorbar and it increases for the graphs from top to bottom. The mass ratio chosen here corresponds to Li-Cs and $n_0 \sigma_g^{3/2} = 1.25 \ 10^{-5}$. The three subfigures correspond to three different values of the interboson scattering length from left to right given by $a_B = 0.7$, 0.9 and 1.1 $L_U$, respectively. Here $L_U = 4.5\sqrt{\sigma_g}$.

the Landau model, because a phase transition in the thermodynamic sense happens when the ground state of the system changes its character. This would be at the point where any cluster becomes lower in energy than the polaron for the first time. The metastability of the polaron can instead be interpreted as a form of hysteresis. The point of the true phase transition is not properly described in the energy functional (35) and requires to improve the model further as we will discuss in the following.

At the point of instability, both the first and second derivative of the energy with respect to $\phi$ vanish and we can use this to find the density and scattering length determining the "stability boundary" of the polaron state. This critical density, as a function of the scattering length, is given by

$$
\begin{aligned}
n_{0,c} &= \frac{T_0 \mu_r (a_{\text{eff}} - a_{\text{eff},-})}{27\pi S_0 a_{\text{eff}}^2 (1 - \frac{2\pi V_0^2 a_{\text{eff}}}{\mu_r T_0})} \\
&= \frac{a_{\text{eff}} - a_{\text{eff},-}}{360\pi a_{\text{eff}}^2 \sigma_g (1 - \frac{5^{5/2} a_{\text{eff}}}{3^4 \sqrt{2\pi\sigma_g}})}.
\end{aligned}
\tag{36}
$$

Finding the inverse equation for $a_{\text{eff}}$ as a function of $n_0$ can be done by solving a third-order polynomial, yielding an analytical, but lengthy expression. Remarkably, the only dependence on the mass in equation Eq. (36) is via $a_{\text{eff},-}$, see Eq. (33).

## 6.3 Including interboson repulsion

To obtain an even clearer analogy to the theory of phase transitions, we now include the interboson repulsion to stabilize the cluster states. The only interboson repulsion term which qualitatively matters for the description of the behaviour at strong coupling is the quartic term in the Hamiltonian of Eq. (4). The quadratic and cubic term are mostly important to determine the shape of the polaron cloud at long distances, which we do not attempt to describe in this

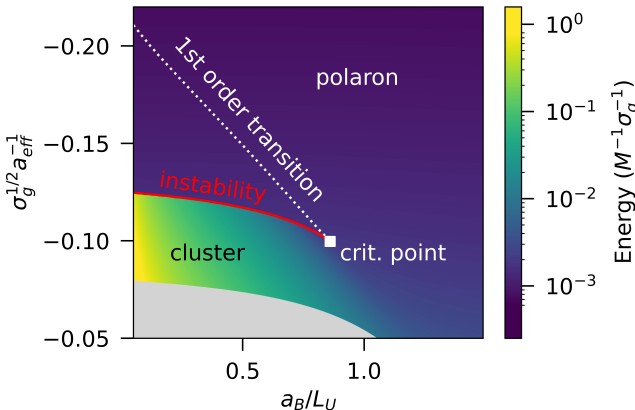

Figure 11: Colormap of the energy of the polaron as a function of the effective impurity-boson scattering length $a_{\text{eff}}$ and the interboson repulsion scattering length $a_B$ as obtained from our analytical model. The density $n_0\sigma_g^{3/2} = 2.5 \ 10^{-5}$ and $L_U = 2\sigma_g^{1/2}$. The red line indicates the polaronic instability and the white dashed line indicates the point where the first-order phase transition happens, where the cluster energy crosses below the polaron energy. Both of these lines end at the critical point. In the grey region in the bottom left, our analytical model is not applicable.

simplified model. Therefore, we only keep the quartic term. Note that this is the opposite approach compared to what is usually done in the Bogoliubov approximation.

To see all the relevant effects we need to expand the energy functional to the third instead of only second order in $\phi^2$, $\xi_0$, and $\xi_1$. At this point we do not optimize the exponents of the Gaussian basis functions again, but simply use the optimized exponents obtained in the previous step. This yields the energy functional

$$E(\phi,\xi_0,\xi_1) = gn_0 + 2gV_0\sqrt{n_0}\phi + (T_0 + gV_0^2)\phi^2 + \frac{U_{00}}{2}\phi^4 + \Big(\frac{Y_0}{2} + 2S_0U_{00}\phi^2\Big)\xi_0^2$$
$$+ \Big[\frac{3Y_1}{2} + 3S_1(U'_{10}\phi^2 + T_L^2\phi^2)\Big]\xi_1^2 + U_{00}\phi^2\xi_0 - 3Y_L\phi^2\xi_1 - 3Y_L\xi_0\xi_1. \tag{37}$$

Here we have defined

$$Y_0 = 2S_0(T_0 + gV_0^2) + U_{00}, \tag{38}$$
$$Y_1 = 2S_1T_1 + 3U_{11}, \tag{39}$$
$$Y_L = T_L^2 - U_{10}. \tag{40}$$

The numbers $U_{00}$, $U_{10}$, $U'_{10}$ and $U_{11}$ are defined in the Appendix in Eqs. (119- 122).

Minimizing Eq. (37) with respect to $\xi_0$ and $\xi_1$ gives

$$\xi_0 = \frac{\big[3Y_L^2 - [Y_1 + 2S_1(U'_{10} + T_L^2)\phi^2]U_{00}\big]\phi^2}{[Y_0 + 4S_0U_{00}\phi^2][Y_1 + 2S_1(U'_{10} + T_L^2)\phi^2] - 3Y_L^2}, \tag{41}$$

$$\xi_1 = \frac{[Y_0 + 4S_0U_{00}\phi^2]\xi_0 + U_{00}\phi^2}{3Y_L}. \tag{42}$$

Taking the limit $\phi \to 0$, we can find a new value for $a_{\text{eff},-}$ as a function of $U$,

$$a_{\text{eff},-}(U) = \frac{T_0\mu}{V_0^2 2\pi}(1 - \frac{2S_0T_0Y_1}{3Y_L^2 - Y_1U_{00}}). \tag{43}$$

When we again expand the energy functional up to quartic order in $\phi$ we retrieve Eq. (35) but with $a_{\text{eff},-}$ replaced by $a_{\text{eff},-}(U)$. To recover the effect of the stabilization of the polaronic collapse, we need to go to higher order in $\phi$. We find

$$E(\phi) = g n_0 + 2 g V_0 \sqrt{n_0} \phi + (T_0 + g V_0^2) \phi^2 - (T_0 + g V_0^2 + 2 U_{00} \phi^2) \xi_0 \phi^2. \qquad (44)$$

Since $\xi_0$ has $\phi^2$ in multiple arguments of the denominator and the numerator, this does not allow a simple expression in terms of $a_{\text{eff},-}(U)$.

Note further, that the quartic interboson repulsion term $\frac{U_{00} \phi^4}{2}$ from Eq. (37) has been absorbed into the quartic term originating from the Gaussian part of the state.

In Fig. 10 the value of the energy functional of Eq. (44) is plotted as a function of the order parameter $\phi \sqrt{S_0}$ for several boson-impurity (indicated via the color of the lines) and interboson scattering lengths.

For weak repulsion [Fig. 10a)] we find a double well picture of a shallow well corresponding to the polaron and a deeper well corresponding to the Efimov cluster. Quantitatively, the cluster has a much weaker binding energy than in the original model, mainly because the exponents of the Gaussian basis functions are optimized for the polaron and not the clusters.

Qualitatively, the physics is very similar as for the full model. In Fig. 10a), the first-order transition is clearly apparent. It occurs when the well corresponding to the cluster on the right becomes lower in energy than the well associated with the polaron state. However, the polaron first remains a metastable local minimum, even though the cluster state is lower in energy. Then, at some critical scattering length, the barrier protecting the polaron disappears and there is a sudden transition from the polaron into the cluster state. This can still be interpreted as a form of polaronic instability.

If the interboson repulsion is increased, the double well picture no longer applies. In Fig. 10b) the energy functional is shown close to the critical point. Here we see that there is no first-order transition or polaronic instability any more, but still the ground state changes rapidly in character for some critical scattering length. Finally, for even stronger repulsion, in Fig. 10c), no sign of a transition remains, and we are deeply in the smooth crossover regime.

In Fig. 11 we capture this behaviour in a single figure, by showing the polaron energy as a function of the boson-impurity and boson-boson scattering length. Indeed, the figure is remarkably similar to Fig. 6c), showing how well our analytical model captures this behaviour. Note that in the grey area in the bottom left, $a_{\text{eff}} < a_{\text{eff},-}(U)$. Here our model breaks down. In particular, in this figure we clearly see the line of first-order transition, where the polaron ceases to be the ground state, and the line of instability, where the polaron becomes completely unstable. These two points merge in the critical point, where the energy functional takes the form as shown in Fig. 10b). At this critical point, where the line of first-order transition terminates, the phase transition turns into a second-order one.

Fig. 11 as a whole, as well as Fig. 6b) and c), are remarkably reminiscent to the phase diagram corresponding to the gas-liquid phase transition. In this analogy the polaron state corresponds to the gaseous state and the cluster state to the liquid state. The gas-liquid transition is also a first-order phase transition, up to a critical point, after which the state is called a supercritical fluid. In the regime where the polaron state is metastable, this would be analogous to a supercooled gas: a gas cooled below its condensation point. Note, however, that in the polaron case, this phase diagram is realized in the quantum regime at zero temperature.

As a final result, in Fig. 12 we plot the "Bose polaron phase diagram" as a function of the mass ratio and the interboson repulsion, based on our analytical model Eq. (44). We see that the stability diagram is remarkably similar to Figs. 1 and 7, even on a quantitative level [3]. This

---

[3] Note for direct comparison that the potential in the analytical model is best compared to the Gaussian potential with $L_g = 2\sigma_g^{1/2}$. This means that the density of Fig. 12 is best compared to the case of Fig. 7a) (note also the difference in the scale of the colormap)

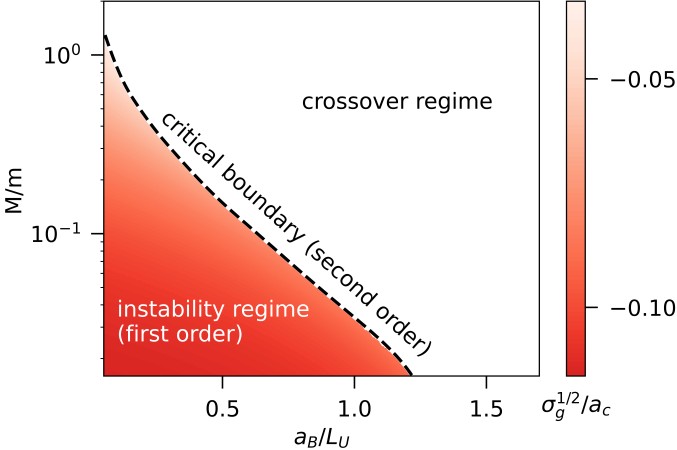

Figure 12: Stability diagram of the polaron as a function of the impurity-boson mass ratio $M/m$ and the interboson repulsive scattering length $a_B$, as obtained from our analytical model. The density $n_0 \sigma_g^{3/2} = 1.25 \ 10^{-5}$ and $L_U = 2\sigma_g^{1/2}$. In the red area of the diagram the polaron undergoes an instability as the boson-impurity interaction strength is swept across a Feshbach resonance. The critical scattering length $a_c$ of the instability is indicated by the colormap. In the white regime the polaron undergoes a smooth crossover instead.

shows that the form of the Bose-polaron phase diagram is remarkably robust.

## 7 Conclusion and Outlook

We have characterized the behaviour of the attractive Bose polaron, i.e., the ground state of a mobile impurity interacting with a Bose-Einstein condensate, across a large parameter regime of boson-impurity and boson-boson scattering lengths, BEC densities and impurity-to-boson mass ratios. Thus, we have brought together the qualitatively different results from several studies into a single, unified theoretical picture. To this end, we have compared two state-of-the-art variational methods: the Gaussian-state and double-excitation approaches. We developed a computationally efficient technique by expressing the variational functions in terms of a set of spherical Gaussian basis functions.

We found that the polaron experiences an instability as predicted in Refs. [70, 71] for weak repulsion and light impurities. This instability turns into a smooth crossover for larger repulsion or heavier impurities. Most of the experiments will naturally be in the crossover regime, but the instability regime should be experimentally accessible with light impurities in BECs with a small interboson scattering length.

We developed a simple analytical model capturing the phenomenology of the instability and crossover. From this model it becomes apparent that the physics of attractive Bose polarons can be understood in the language of the Landau model of first-order phase transitions. It is in fact strongly reminiscent of the gas-liquid phase transition, where the polaron state is analogous to the gaseous phase and the cluster state to the liquid phase. Clearly, in the case of a single impurity we cannot truly speak of a phase transition in our model, but it is likely that this will turn into a proper quantum phase transition when considering a finite density of impurities. We have furthermore shown that the stability diagram of the polaron is reproduced surprisingly well by the simple analytical model. A topic inviting further study would be a more detailed characterization of the critical behaviour at the critical point where the phase

transition turns second order.

Given that the properties of the ground state differ strongly across the parameter regimes, an interesting question remains how these effects would manifest themselves in the dynamics and the excitation spectrum of the polaron. Especially interesting would be to see whether some resonant behaviour occurs at the onset of the polaronic instability, in analogy to the Efimov resonance [70, 71]. Dynamical calculations have so far been carried out mostly using (extended) Gross-Pitaevskii equations [103] or equivalently, with a coherent state approach [74, 104, 105]. To see the effects mentioned above one would need to explicitly allow for the correlations between the bosons induced by the impurity. In principle, the Gaussian-state approach can also be applied for real-time evolution, but this is computationally more challenging due to the more oscillatory nature of the wave function. For this reason, the parameterization of the variational functions used here in terms of Gaussian basis functions, is likely not optimal for real-time evolution.

Furthermore, also interesting effects are predicted to appear in the finite temperature behaviour of the Bose polaron [106–109], although several approaches have again found conflicting results. Therefore, it would be a promising avenue of study to see whether a unified model can also be developed capturing the finite-temperature properties for varying mass ratios and background repulsion.

In the context of Bose-Fermi mixtures, mediated interactions by fermions in Bose-Einstein condensates have been studied already both theoretically [110–112] and experimentally [32, 33] in a regime of finite fermion density. However, in our work the fermion-mediated interactions are caused by a different mechanism. Namely, they arise from the Efimov effect, instead of the many-body effects originating from the Fermi surface of the fermions. Since we show that the impurity-mediated interactions play a profound role in the Bose polaron model, it can be expected it will also be important in the general description of finite-density mixtures at strong interactions. We hope our work can provide a starting point for fruitful studies in this direction. In particular, we hope that further inspiration can be drawn from quantum chemistry to also tackle the bipolaron problem.

Bose polarons have recently been observed for the first time in two-dimensional semiconductors coupled to a microcavity [113]. Even though the standard Efimov effect does not exist in two dimensions, it is still possible that in certain conditions bound states with more than two particles can form. Therefore, similar phenomena as we have predicted here in the context of cold atomic gases might well be found in these solid state platforms.

*Note added.* During the preparation of the manuscript a related work has appeared [114]. Here the authors introduce another variational approach to the Bose polaron problem, which allows to extract excited states in the spectrum. However, the authors include only limited interboson correlations, meaning that the Efimov effect, which is underlying most of our results, is not captured.

## Acknowledgements

We thank Martin Zwierlein and Carsten Robens for inspiring discussions. R.S. acknowledges support by the Deutsche Forschungsgemeinschaft under Germany's Excellence Strategy EXC 2181/1-390900948 (the Heidelberg STRUCTURES Excellence Cluster). J.I.C. thanks the Munich Center for Quantum Science and Technology for support.

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

# A   Gaussian states and Gaussian basis

Let us denote our variational Gaussian-state Ansatz as follows

$$|GS\rangle = \hat{V}|0\rangle = \hat{U}_\phi \hat{V}_\xi |0\rangle = \mathcal{N} \exp[\int_k (\phi(k)\hat{b}_k^\dagger - \phi^*(k)\hat{b}_k)] \exp[\frac{1}{2}\int_k \int_{k'} \hat{b}_k^\dagger \xi(k,k')\hat{b}_{k'}^\dagger]|0\rangle. \tag{45}$$

It is given by a unitary displacement operator and a squeezing operator acting on the vacuum. These operators are respectively characterized by the coherent displacement $\phi$ and the symmetric correlation function $\xi$. Often also the squeezing operator is written as a unitary operator [76], but we have chosen the non-unitary form containing only creation operators because this is simpler to work with for our purposes. Note that any Gaussian state can be written in this form by normal ordering the Gaussian unitary [115], so that the terms with annihilation operators disappear when acting on the vacuum. The commutators of $\hat{V}$ with the bosonic creation and annihilation operators are given by

$$[\hat{b}_k^\dagger, \hat{V}] = \phi^*(k), \tag{46}$$

$$[\hat{b}_k, \hat{V}] = \phi(k) + \int_{k'} \xi(k,k')\hat{b}_{k'}^\dagger. \tag{47}$$

Now we parameterize our coherent and Gaussian variational functions as

$$\phi(k) = \sum_i \phi_i \chi_{00}(\sigma_i^{(\phi)}, k), \tag{48}$$

$$\xi(k,k') = \sum_{lij} \sum_{m=-l}^{l} (-1)^m \xi_{ij}^{(l)} \chi_{lm}(\sigma_i^{(\xi,l)}, k) \chi_{l-m}(\sigma_j^{(\xi,l)}, k). \tag{49}$$

Here the functions $\chi_{lm}(\sigma, k)$ are spherical Gaussian basis functions,

$$\chi_{lm}(\sigma, k) = (2\pi)^{3/2} Y_{lm}(\theta, \phi) i^{-l} k^l \exp(-\sigma k^2). \tag{50}$$

Because our model has spherical symmetry, $\xi_{ij}^{(l)}$ is only dependent on $l$ and not on $m$. The basis functions are not orthonormal and we define the Hermitian overlap matrix $S$ as

$$S_{ij}^{(l)} = \int_k \chi_{lm}^*(\sigma_i^{(\xi,l)}, k) \chi_{lm}(\sigma_j^{(\xi,l)}, k) = \frac{\sqrt{\pi}(2l+1)!!}{2^{l+2}(\sigma_i^{(\xi,l)} + \sigma_j^{(\xi,l)})^{3/2+l}}. \tag{51}$$

In a similar way, we define $S^{(\phi)}$, where $l, m = 0$ and the exponents correspond to $\sigma^{(\phi)}$, and we define $S^{(mix)}$

$$S_{ij}^{(mix)} = \int_k \chi_{00}^*(\sigma_i^{(\phi)}, k) \chi_{00}(\sigma_j^{(\xi,0)}, k). \tag{52}$$

One can view $\xi$ and $S$ as block diagonal matrices where the blocks are labeled by $(l)$.

We will now show how to compute expectation values with respect to our variational state. For a single annihilation operator (analogous for the creation operator) one finds

$$\langle GS|\hat{b}_k|GS\rangle = \phi(k) = \phi\chi_\phi(k) \tag{53}$$

Here $\chi_\phi$ is a vector of the coherent state basis functions. For the two-point functions this is slightly more involved:

$$\langle GS|\hat{b}_{k'}^\dagger \hat{b}_r|GS\rangle = \phi(k)\phi^*(k') + \langle 0|\hat{V}_\xi^\dagger \hat{b}_{k'}^\dagger \hat{b}_k \hat{V}_\xi|0\rangle. \tag{54}$$

For the expectation value with respect to the squeezed state we can derive the expression recursively,

$$G(\boldsymbol{k}, \boldsymbol{k}') = \langle 0|\hat{V}_\xi^\dagger \hat{b}_{\boldsymbol{k}'}^\dagger \hat{b}_{\boldsymbol{k}} \hat{V}_\xi|0\rangle = \int_q \int_{q'} \xi^*(\boldsymbol{k}', \boldsymbol{q}')[\delta(\boldsymbol{q}', \boldsymbol{q}) + G(\boldsymbol{q}', \boldsymbol{q})]\xi(\boldsymbol{q}, \boldsymbol{k}), \tag{55}$$

We now define

$$G(\boldsymbol{k}, \boldsymbol{k}') = \sum_l \sum_{m=-l}^{l} \chi_{lm}^T(\boldsymbol{k}) G^{(l)} \chi_{lm}^*(\boldsymbol{k}'), \tag{56}$$

where $G$ is again a block diagonal matrix with blocks labeled by $l$, and $\chi_{lm}$ is the vector of Gaussian-state basis functions with angular momentum labels $l$ and $m$. With this notation, Eq. (60) can be cast into matrix form:

$$G = \xi S \xi^* + \xi G^T \xi^*. \tag{57}$$

Eq. 57 can be solved in terms of a geometric series:

$$G = \sum_{n=1}^{N} (\xi S^T \xi^* S)^n S^{-1} = \left[ [\mathbb{1} - (\xi S^T \xi^* S)]^{-1} - \mathbb{1} \right] S^{-1} \tag{58}$$

$$\equiv X - S^{-1}. \tag{59}$$

We can follow a similar procedure for the other type of correlation functions:

$$F(\boldsymbol{k}, \boldsymbol{k}') = \sum_l \sum_{m=-l}^{l} \chi_{lm}^T(\boldsymbol{k}) F^{(l)} \chi_{l-m}(\boldsymbol{k}') \langle 0|\hat{V}_\xi^\dagger \hat{b}_{\boldsymbol{k}} \hat{b}_{\boldsymbol{k}'} \hat{V}_\xi|0\rangle. \tag{60}$$

Again, one can recursively find an expression for $F$:

$$F = XS\xi = \xi S^T X^T = \xi + GS\xi. \tag{61}$$

Higher-order correlation functions can now be computed employing Wick's theorem, using the expressions for the displacement and the two-point functions.

Finally, the norm of the variational state can be written as

$$\langle GS|GS \rangle = \frac{|\mathcal{N}|^2}{\sqrt{\prod_l \det[(\mathbb{1} - S^{(l)}\xi^{(l)} S^{(l)}[\xi^{(l)}]^*)]^{(2l+1)}}}. \tag{62}$$

# B    Imaginary time evolution

We optimize the energy of our state using imaginary time evolution. Here we derive the equations of motion. We use McLachlan's variational principle

$$\frac{\partial}{\partial(\partial_\tau x_j)} ||\partial_\tau + \hat{\mathcal{H}} - E|\psi(\boldsymbol{x})\rangle|^2 = 0, \tag{63}$$

where $E$ is the energy, given as

$$\langle \psi(\boldsymbol{x})|\hat{\mathcal{H}}|\psi(\boldsymbol{x})\rangle = E\langle \psi(\boldsymbol{x})|\psi(\boldsymbol{x})\rangle, \tag{64}$$

where we assume the norm of the state to be 1. The energy term is needed in Eq. (63) to ensure conservation of the norm. Now the derivative $\partial_\tau$ can be replaced by:

$$\partial_\tau = \sum_i \partial_\tau x_i \partial_{x_i}. \tag{65}$$

Inserting this into Eq. (63) gives:

$$\partial_{x_j} E + \sum_i \partial_\tau x_i \big[ \langle \psi(\boldsymbol{x}) | \overleftarrow{\partial_{x_i}} \partial_{x_j} | \psi(\boldsymbol{x}) \rangle + \langle \psi(\boldsymbol{x}) | \overleftarrow{\partial_{x_j}} \partial_{x_i} | \psi(\boldsymbol{x}) \rangle \big] = 0. \tag{66}$$

The parameter vector $\boldsymbol{x}$ consists of $\mathcal{N}$, $\mathcal{N}^*$, $\phi$, $\phi^*$, $\xi$ and $\xi^*$. The derivatives $\partial_{x_i} | \psi \rangle$ are given by:

$$\partial_{\mathcal{N}} | \psi \rangle = \frac{1}{\mathcal{N}} | \psi \rangle, \tag{67}$$

$$\partial_{\mathcal{N}^*} | \psi \rangle = 0, \tag{68}$$

$$\partial_{\phi_i} | \psi \rangle = \int_{\boldsymbol{k}} \chi_{00}(\sigma_{\phi,i}, \boldsymbol{k}) [\hat{b}_{\boldsymbol{k}}^\dagger - \frac{\phi^*(\boldsymbol{k})}{2}] | \psi \rangle, \tag{69}$$

$$\partial_{\phi_i^*} | \psi \rangle = \int_{\boldsymbol{k}} \chi_{00}(\sigma_{\phi,i}, \boldsymbol{k}) [-\hat{b}_{\boldsymbol{k}} + \frac{\phi(\boldsymbol{k})}{2}] | \psi \rangle, \tag{70}$$

$$\frac{\partial}{\partial \xi_{ij}^{(l)}} | \psi \rangle = \hat{U}_\phi \sum_{m=-l}^{l} (-1)^m \int_{\boldsymbol{k}, \boldsymbol{k}'} \chi_{lm}(\sigma_{\xi,li}, \boldsymbol{k}) \chi_{l-m}(\sigma_{\xi,lj}, \boldsymbol{k}') \hat{b}_{\boldsymbol{k}}^\dagger \hat{b}_{\boldsymbol{k}'}^\dagger \hat{V}_\xi | 0 \rangle, \tag{71}$$

$$\frac{\partial}{\partial [\xi_{ij}^{(l)}]^*} | \psi \rangle = 0. \tag{72}$$

Let us now denote $\boldsymbol{x}'$ as the vector $\boldsymbol{x}$ without $\mathcal{N}$ and $\mathcal{N}^*$. If we take $x_j = \mathcal{N}$ in Eq. (66) we arrive at

$$0 = \frac{\partial_\tau \mathcal{N}^*}{|\mathcal{N}|^2} + \frac{1}{\mathcal{N}} \sum_i \partial_\tau x_i' \langle \psi(\boldsymbol{x}) | \overleftarrow{\partial_{x_i'}} | \psi(\boldsymbol{x}) \rangle. \tag{73}$$

Multiplying this equation by $|\mathcal{N}|^2 \mathcal{N}$ and adding it to the complex conjugate, we exactly recover the equation for the norm

$$0 = \mathcal{N} \partial_\tau \mathcal{N}^* + \mathcal{N}^* \partial_\tau \mathcal{N} + |\mathcal{N}|^2 \sum_i \partial_\tau x_i' \big[ \langle \psi(\boldsymbol{x}) | \overleftarrow{\partial_{x_i'}} | \psi(\boldsymbol{x}) \rangle + \langle \psi(\boldsymbol{x}) | \partial_{x_i'} | \psi(\boldsymbol{x}) \rangle \big] = \partial_\tau \langle \psi(\boldsymbol{x}) | \psi(\boldsymbol{x}) \rangle. \tag{74}$$

This shows that indeed the norm is conserved. For the other equations:

$$0 = \partial_{x_j'} E + \frac{\partial_\tau \mathcal{N}}{\mathcal{N}} \langle \psi(\boldsymbol{x}) | \overleftarrow{\partial_{x_j'}} | \psi(\boldsymbol{x}) \rangle + \frac{\partial_\tau \mathcal{N}^*}{\mathcal{N}^*} \langle \psi(\boldsymbol{x}) | \partial_{x_j'} | \psi(\boldsymbol{x}) \rangle$$
$$+ \sum_i \partial_\tau x_i' \big[ \langle \psi(\boldsymbol{x}) | \overleftarrow{\partial_{x_i'}} \partial_{x_j'} | \psi(\boldsymbol{x}) \rangle + \langle \psi(\boldsymbol{x}) | \overleftarrow{\partial_{x_j'}} \partial_{x_i'} | \psi(\boldsymbol{x}) \rangle \big]. \tag{75}$$

Inserting now Eq. 73 gives:

$$0 = \partial_{x_j'} E + \sum_i \partial_\tau x_i' \big[ \langle \psi(\boldsymbol{x}) | \overleftarrow{\partial_{x_i'}} \partial_{x_j'} | \psi(\boldsymbol{x}) \rangle - \langle \psi(\boldsymbol{x}) | \overleftarrow{\partial_{x_i'}} | \psi(\boldsymbol{x}) \rangle \langle \psi(\boldsymbol{x}) | \partial_{x_j'} | \psi(\boldsymbol{x}) \rangle$$
$$+ \langle \psi(\boldsymbol{x}) | \overleftarrow{\partial_{x_j'}} \partial_{x_i'} | \psi(\boldsymbol{x}) \rangle - \langle \psi(\boldsymbol{x}) | \overleftarrow{\partial_{x_j'}} | \psi(\boldsymbol{x}) \rangle \langle \psi(\boldsymbol{x}) | \partial_{x_i'} | \psi(\boldsymbol{x}) \rangle \big]. \tag{76}$$

For $x_j' = \phi_j$ we find

$$0 = \eta_j^* + \sum_i \partial_\tau \phi_i^* [S_{ij}^{(\phi)} + 2(S^{(mix)} G^{(0)} [S^{(mix)}]^T)_{ij}] - \sum_i \partial_\tau \phi_i [2(S^{(mix)} [F^{(0)}]^* [S^{(mix)}]^T)_{ij}], \tag{77}$$

where $\eta_j^* = \partial_{\phi_j} E$. In practice, taking a simplified equation for $\phi$

$$\partial_\tau \phi = -[S^{(\phi)}]^{-1} \boldsymbol{\eta}, \tag{78}$$

gives the same result with comparable computational efficiency.

For $x'_j = \xi_{qp}$ we get

$$0 = \partial_{\xi_{qp}} E + \sum_{ij} \partial_\tau \xi^*_{ij} [S\Lambda XS]_{iq} [SXS]_{jp}, \tag{79}$$

where $\Lambda$ is a block-diagonal matrix with $\Lambda^{(l)} = (2l+1)\mathbb{1}$.

The energy $E$ is most simply expressed in terms of $F$ and $G$ because of Wick's theorem. If we thus define

$$(\Lambda\mathcal{E})_{ij} = \frac{\partial E}{\partial G_{ji}}, \tag{80}$$

$$(\Lambda\Delta)_{ij} = 2\frac{\partial E}{\partial F^*_{ij}}, \tag{81}$$

we can express these derivatives in terms of $\xi$ via

$$\frac{\partial X_{ij}}{\partial \xi_{mn}} = (XS)_{im}(S^T\xi^*SX)_{nj} = (XS)_{im}(S^TF^*)_{nj}, \tag{82}$$

$$\frac{\partial X_{ij}}{\partial \xi^*_{mn}} = (XS\xi S^T)_{im}(SX)_{nj} = (FS^T)_{im}(SX)_{nj}. \tag{83}$$

This yields the total expression

$$\frac{\partial E}{\partial \xi} = \frac{\Lambda}{2}[S^TX^T\mathcal{E}^TF^*S + S^TF^*\mathcal{E}XS + S^TX^T\Delta^*XS + S^TF^*\Delta F^*S]. \tag{84}$$

Inserting this into Ref. 79 and multiplying with $(SXS)^{-1}$ from the left (transposed) and right, yields

$$\partial_\tau \xi^* = -[S^{-1}\Delta^*S^{-1} + S^{-1}\mathcal{E}^T\xi^* + \xi^*\mathcal{E}S^{-1} + \xi^*\Delta\xi^*]. \tag{85}$$

This derivation can straightforwardly be applied also to the case of real-time evolution. The equations of motion for $\xi$ for real-time evolution are also given for a general case in Appendix G of Ref. [76]. In other works using the Gaussian-state variational approach the equations of motion are usually formulated in terms of the covariance matrix $\Gamma$ [71, 76, 116]. These equations of motion can be extracted directly from the equations for $\xi$ and $\xi^*$. However, we find that for imaginary time evolution, it is numerically more stable to work with $\xi$ instead of $\Gamma$.

## C   Treatment of the interboson repulsion

The interboson repulsion term of the Hamiltonian is given by

$$\hat{\mathcal{H}}_U = \int_r \int_{r'} V_{BB}(r'-r)\Big[\frac{n_0}{2}(2\hat{b}^\dagger_{r'}\hat{b}_r + \hat{b}^\dagger_{r'}\hat{b}^\dagger_r + \hat{b}_{r'}\hat{b}_r) + \sqrt{n_0}(\hat{b}^\dagger_{r'}\hat{b}^\dagger_r\hat{b}_r + \hat{b}^\dagger_r\hat{b}_{r'}\hat{b}_r) + \frac{1}{2}\hat{b}^\dagger_{r'}\hat{b}^\dagger_r\hat{b}_{r'}\hat{b}_r\Big]. \tag{86}$$

The Gaussian-state expectation value with respect to this term is given by

$$
\begin{aligned}
E_U(\phi,\xi) &= \langle GS|\hat{\mathcal{H}}_U|GS\rangle \\
&= \frac{U}{2L_U^2}\int_r\int_{r'}\exp\Big[-\frac{(r'-r)^2}{L_U^2}\Big]\Big\{\frac{n_0}{2}\Big[\phi^*(r)\phi(r')+\phi(r)\phi(r')+F(r',r)+G(r',r)+h.c.\Big] \\
&\quad+\sqrt{n_0}\Big[|\phi(r)|^2\phi(r')+G(r,r)\phi(r')+G(r',r)\phi(r)+F(r',r)\phi^*(r)+h.c.\Big] \\
&\quad+\frac{1}{2}\Big[|\phi(r)|^2|\phi(r')|^2+\big[|\phi(r')|^2+\frac{1}{2}G(r',r')\big]G(r,r) \\
&\quad+\big[\phi^*(r)\phi(r')+\frac{1}{2}G(r',r)\big]G(r,r')+\big[\phi^*(r)\phi^*(r')+\frac{1}{2}F^*(r,r')\big]F(r,r')+h.c.\Big]\Big\}. \quad (87)
\end{aligned}
$$

For the ground state scenario, $\phi$ is real, meaning that $\phi^*=\phi$. The same holds for $F$ and $G$. Note that the $F$-terms typically dominate over the $G$-terms in the polaron regime, since $F\sim\xi$ and $G\sim\xi^2$, and $\xi$ is generally small. Furthermore, $F$ is usually negative, so that the $\phi^2 F^*$ terms give a negative energy contribution, whereas $G$ is always positive.

We approximate Eq. (87) by:

$$
\begin{aligned}
E_U(\phi,\xi) &\approx \frac{1}{2L_U^2}\int_r\int_{r'}\exp\Big[-\frac{(r'-r)^2}{L_U^2}\Big]\Big\{\frac{n_0 U_B}{2}\Big[\phi^*(r)\phi(r')+\phi(r)\phi(r')+G(r',r)+h.c.\Big] \\
&\quad+\sqrt{n_0}U_B\Big[|\phi(r)|^2\phi(r')+G(r,r)\phi(r')+G(r',r)\phi(r)+h.c.\Big] \\
&\quad+\frac{U}{2}\Big[|\phi(r)|^2|\phi(r')|^2+\big[|\phi(r')|^2+\frac{1}{2}G(r',r')\big]G(r,r) \\
&\quad+\big[(\phi^*(r)\phi(r')+\frac{1}{2}G(r',r)\big]G(r,r')+\big[\phi^*(r)\phi^*(r')+\frac{1}{2}F^*(r,r')\big]F(r,r')+h.c.\Big]\Big\}, \quad (88)
\end{aligned}
$$

where the coupling $U_B=\frac{8a_B}{m\sqrt{\pi}L_U}$ yields the correct scattering length on the level of the Born approximation. Note that in the quadratic and cubic terms $U$ is replaced by $U_B$ and the terms involving $F$ are dropped. The terms involving $F$ are the ones which actually renormalize the interactions, so these need to be removed when taking the Born approximation. The terms involving $G$ are only of minor importance and describe the repulsive interactions created by the excitations originating from the Gaussian part of the wave function, which for example lead to the Lee-Huang-Yang-correction [117].

For the double-excitation Ansatz

$$
\hat{A}[\beta_0,\beta(k),\alpha(k,k')] = \beta_0 + \int_k\beta(k)\hat{b}_k^\dagger + \frac{1}{\sqrt{2}}\int_k\int_{k'}\alpha(k,k')\hat{b}_k^\dagger\hat{b}_{k'}^\dagger, \quad (89)
$$

the value of $E_U(\beta_0,\beta,\alpha)=\langle DE|\hat{\mathcal{H}}_U|DE\rangle$ is given by

$$
\begin{aligned}
E_U(\beta_0,\beta,\alpha) &= \frac{U}{2L_U^2}\int_r\int_{r'}\exp\Big[-\frac{(r'-r)^2}{L_U^2}\Big]\Big\{\frac{n_0}{2}\big[\beta^*(r)\beta(r')+\sqrt{2}\beta_0^*\alpha(r',r)+ \\
&\quad 2\int_{r''}\alpha^*(r',r'')\alpha(r'',r)+h.c.\big]+\sqrt{2n_0}\big[\beta^*(r)\alpha(r,r')+h.c.\big]+|\alpha(r',r)|^2\Big\}. \quad (90)
\end{aligned}
$$

In this case it is much more ambiguous how to take the Born approximation since this is not a mean field approach and the best strategy depends on whether we are in the polaronic or molecular regime. To be most consistent with the Gaussian-state case we choose to use the

following energy functional

$$
E_U(\beta_0, \beta, \alpha) \approx \frac{1}{2L_U^2} \int_r \int_{r'} \exp\left[-\frac{(r'-r)^2}{L_U^2}\right]\left\{\frac{n_0 U_B}{2}\left[2\beta^*(r)\beta(r')\right.\right.
$$

$$
\left.\left. + 4\int_{r''} \alpha^*(r', r'')\alpha(r'', r) + h.c.\right] + \sqrt{2n_0}U_B\left[\beta^*(r)\alpha(r, r') + h.c.\right] + U|\alpha(r', r)|^2\right\}. \quad (91)
$$

This energy functional is chosen to reproduce the mean field result in the weak coupling limit. The quartic term is still described exactly, which is most important for our work.

# D  Computation of Gaussian integrals

The expectation values with respect to the kinetic, boson-impurity interaction and LLP-terms yield relatively simple expressions.

For the kinetic energy we find

$$
T_{l,ij} = \int_k \frac{k^2}{2\mu_r} \chi_{lm}^*(\sigma_i, k)\chi_{lm}(\sigma_j, k) = \frac{\sqrt{\pi}(2l+3)!!}{2^{l+4}\mu_r(\sigma_{l,i} + \sigma_{l,j})^{5/2+l}}. \quad (92)
$$

For the Gaussian-state expectation values of the LLP-term we find terms of the form

$$
T_{L,l,ijpq} = -\sum_{m'} \frac{(-1)^{m+m'}}{M}\left[\int_k k\chi_{(l+1)m'}^*(\sigma_i, k)\chi_{lm}(\sigma_j, k)\right]\left[\int_{k'} k'\chi_{(l+1)-m'}^*(\sigma_p, k')\chi_{l-m}(\sigma_q, k')\right], \quad (93)
$$

$$
= \frac{1}{M}\sum_{m'}\left[\int_k k\chi_{(l+1)m'}^*(\sigma_i, k)\chi_{lm}(\sigma_j, k)\right]\cdot\left[\int_{k'} k'\chi_{lm}^*(\sigma_q, k')\chi_{(l+1)m'}(\sigma_p, k')\right], \quad (94)
$$

$$
= \frac{\pi(l+1)[(2l+3)!!]^2}{(2l+1)M2^{2l+6}(\sigma_{l+1,i} + \sigma_{l,j})^{5/2+l}(\sigma_{l+1,p} + \sigma_{l,q})^{5/2+l}}. \quad (95)
$$

For the interaction term with the Gaussian boson-impurity interaction potential (10) it is most convenient to carry out the integral in real space, and define the Fourier transform $\tilde{\chi}$ of the Gaussian orbitals. Then we find

$$
\tilde{\chi}_{lm}(\sigma, r) = \frac{1}{(2\sigma)^{l+3/2}}Y_{lm}(\theta, \phi)\exp(-\frac{r^2}{4\sigma}), \quad (96)
$$

$$
V_{l,i}^{(1)} = \frac{1}{2L_g^2}\int_r \exp(-\frac{r^2}{L_g^2})\tilde{\chi}_{lm}(\sigma_i, r) = \delta_{l,0}\frac{\pi L_g}{2^{1/2}(L_g^2 + 4\sigma_i)^{3/2}}, \quad (97)
$$

$$
V_{l,ij}^{(2)} = \frac{1}{2L_g^2}\int_r \exp(-\frac{r^2}{L_g^2})\tilde{\chi}_{lm}^*(\sigma_i, r)\tilde{\chi}_{lm}(\sigma_j, r) = \frac{\sqrt{\pi}(2l+1)!!L_g^{2l+1}}{2^{l+3}[L_g^2(\sigma_i + \sigma_j) + 4\sigma_i\sigma_j]^{l+3/2}}. \quad (98)
$$

For the separable Gaussian potential used in Sec. 6

$$
V_{l,i} = \int_k \exp(-\sigma_g k^2)\chi_{lm}(\sigma_i, k) = \frac{1}{4\sqrt{2\pi}(\sigma_i + \sigma_g)^{3/2}}. \quad (99)
$$

$$
(100)
$$

Computing the expectation values of the interboson repulsion term is more complex. In this case, we need to calculate integrals of the form (where $j = (n, l, m)$):

$$
I_{j_1, j_2, j_3, j_4} = \int_{r_1} \int_{r_2} r_1^{l_1+l_3} r_2^{l_2+l_4} Y_{l_1 m_1}^*(\Omega_{r_1})Y_{l_3 m_3}(\Omega_{r_1})Y_{l_2 m_2}^*(\Omega_{r_2})Y_{l_4 m_4}(\Omega_{r_2})
$$

$$
\exp(-\alpha_1 r_1^2)\exp(-\alpha_3 r_1^2)\exp(-\alpha_2 r_2^2)\exp(-\alpha_4 r_2^2)\exp\left[-(r_1 - r_2)^2/L_U^2\right]. \quad (101)
$$

Here we use the combined indices $j_1 = (j_1 l_1 m_1)$, where the first index labels the exponent $\alpha_{j_1}$, for which $\alpha_i = \frac{1}{4\sigma_i}$. The integral is nontrivial because the interaction potential depends on the distance between the bosons $(r_1 - r_2)$. To perform the angular and radial integrals we need to write this in a different form. Indeed, following Ref. [118] we can decompose the term according to

$$\exp[-\alpha_U(r_1 - r_2)^2] = \sum_L (2L+1) i_L(2\alpha_U r_1 r_2) \exp[-\alpha_U(r_1^2 + r_2^2)] P_L(\cos(\theta_{12})), \qquad (102)$$

$$= 4\pi \sum_L \sum_M i_L(2\alpha_U r_1 r_2) \exp[-\alpha_U(r_1^2 + r_2^2)] Y_{LM}^*(\mathbf{\Omega}_{r_1}) Y_{LM}(\mathbf{\Omega}_{r_2}), \quad (103)$$

where we defined $\alpha_U = 1/L_U^2$. Here $i_l$ is the modified spherical Bessel function

$$i_L(x) = i^l j_L(ix). \qquad (104)$$

Now we can write the integral (101) as

$$I_{j_1, j_2, j_3, j_4} = \sum_{LM} \int d\Omega_{r_1} Y_{l_1 m_1}^*(\mathbf{\Omega}_{r_1}) Y_{l_3 m_3}(\mathbf{\Omega}_{r_1}) Y_{LM}^*(\mathbf{\Omega}_{r_1}) \int d\Omega_{r_1} Y_{l_2 m_2}^*(\mathbf{\Omega}_{r_2}) Y_{l_4 m_4}(\mathbf{\Omega}_{r_4}) Y_{LM}(\mathbf{\Omega}_{r_4})$$

$$\int dr_1 \, r_1^{2+l_1+l_3} \exp[-(\alpha_1 + \alpha_3 + \alpha_U) r_1^2] \int dr_2 \, r_2^{2+l_2+l_4} \exp[-(\alpha_2 + \alpha_4 + \alpha_U) r_2^2] i_L(2\alpha_U r_1 r_2). \qquad (105)$$

For the angular integrals we use the identity

$$\int d\Omega Y_{l_1 m_1}(\mathbf{\Omega}) Y_{l_2 m_2}(\mathbf{\Omega}) Y_{l_3 m_3}^*(\mathbf{\Omega}) =$$

$$(-1)^{m_3} \sqrt{\frac{(2l_1+1)(2l_2+1)(2l_3+1)}{4\pi}} \begin{pmatrix} l_1 & l_2 & l_3 \\ 0 & 0 & 0 \end{pmatrix} \begin{pmatrix} l_1 & l_2 & l_3 \\ m_1 & m_2 & -m_3 \end{pmatrix}. \quad (106)$$

Without using the symmetries of the problem, these expressions cannot be simplified further.

The radial integrals need to be carried out in two steps, since the Bessel function $i_L$ contains both $r_1$ and $r_2$. In the first step we carry out the following integral

$$\int dr_2 \, r_2^{2+l_2+l_4} \exp[-(\alpha_2 + \alpha_4 + \alpha_U) r_2^2] i_L(2\alpha_U r_1 r_2) =$$

$$\frac{\sqrt{\pi}(2\alpha_U r_1)^L \Gamma(\frac{L+l_2+l_4+3}{2})}{2^{L+2}(\alpha_2+\alpha_4+\alpha_U)^{\frac{(l_2+l_4+L+3)}{2}} \Gamma(L+\frac{3}{2})} \, {}_1F_1\Big(\frac{L+l_2+l_4+3}{2}; L+\frac{3}{2}; \frac{\alpha_U^2 r_1^2}{\alpha_2+\alpha_4+\alpha_U}\Big). \quad (107)$$

Here we used from Ref. [119]

$$\int_0^\infty x^\mu e^{-\alpha x^2} J_\nu(\beta(x)) dx = \frac{\beta^\nu \Gamma(\frac{\nu+\mu+1}{2})}{2^{\nu+1} \alpha^{\frac{1}{2}(\mu+\nu+1)} \Gamma(\nu+1)} \, {}_1F_1\Big(\frac{\nu+\mu+1}{2}; \nu+1; -\frac{\beta^2}{4\alpha}\Big), \qquad (108)$$

where $J_\nu$ is the standard (non-spherical) Bessel function and ${}_1F_1$ is the confluent hypergeometric function. Now the integral that remains is

$$\int dr_1 \, r_1^{2+l_1+l_3+L} \exp[-(\alpha_1+\alpha_3+\alpha_U) r_1^2] {}_1F_1\Big(\frac{L+l_2+l_4+3}{2}; L+\frac{3}{2}; \frac{\alpha_U^2 r_1^2}{\alpha_2+\alpha_4+\alpha_U}\Big). \qquad (109)$$

After replacing the integration variable: $t = r_1^2$ we use another equation from Ref. [119]:

$$\int_0^\infty e^{-st} t^{b-1} \, {}_1F_1(a;c;kt) dt = \Gamma(b) s^{-b} F(a,b;c;ks^{-1}). \tag{110}$$

Here $F(a,b;c;z)$ is the hypergeometric function (sometimes also denoted as ${}_2F_1$). Using this identity gives

$$\frac{1}{2} \int dt \; t^{\frac{1+l_1+l_3+L}{2}} \exp[-(\alpha_1 + \alpha_3 + \alpha_U)t] {}_1F_1\left(\frac{L+l_2+l_4+3}{2}; L + \frac{3}{2}; \frac{\alpha_U^2 t}{\alpha_2 + \alpha_4 + \alpha_U}\right)$$

$$= \frac{\Gamma(\frac{l_1+l_3+L+3}{2})}{2(\alpha_1 + \alpha_3 + \alpha_U)^{\frac{l_1+l_3+L+3}{2}}} F\left(\frac{L+l_2+l_4+3}{2}, \frac{L+l_1+l_3+3}{2}; L + \frac{3}{2}; \frac{\alpha_U^2}{(\alpha_1 + \alpha_3 + \alpha_U)(\alpha_2 + \alpha_4 + \alpha_U)}\right). \tag{111}$$

Putting this together with the prefactor of Eq. (107) and the integral over the angular parts (Eq. (106)) gives

$$I_{j_1,j_2,j_3,j_4} = \frac{\sqrt{\pi}}{8} \sum_{LM} (-1)^{m_2+m_3} \begin{pmatrix} l_3 & l_1 & L \\ 0 & 0 & 0 \end{pmatrix} \begin{pmatrix} l_3 & l_1 & L \\ -m_3 & m_1 & M \end{pmatrix} \begin{pmatrix} l_2 & l_4 & L \\ 0 & 0 & 0 \end{pmatrix} \begin{pmatrix} l_2 & l_4 & L \\ -m_2 & m_4 & M \end{pmatrix}$$

$$\frac{(2L+1)\sqrt{(2l_1+1)(2l_2+1)(2l_3+1)(2l_4+1)} \Gamma(\frac{L+l_2+l_4+3}{2}) \Gamma(\frac{l_1+l_3+L+3}{2}) \alpha_U^L}{\Gamma(L+\frac{3}{2})(\alpha_1 + \alpha_3 + \alpha_U)^{\frac{l_1+l_3+L+3}{2}} (\alpha_2 + \alpha_4 + \alpha_U)^{\frac{(l_2+l_4+L+3)}{2}}}$$

$$F\left(\frac{L+l_2+l_4+3}{2}, \frac{L+l_1+l_3+3}{2}; L + \frac{3}{2}; \frac{\alpha_U^2}{(\alpha_1 + \alpha_3 + \alpha_U)(\alpha_2 + \alpha_4 + \alpha_U)}\right). \tag{112}$$

The expression can be simplified when taking our symmetries into account. For the coherent part for example, $l_1 = l_2 = l_3 = l_4 = 0$. In this case we find

$$I_{\phi^4} = \frac{\sqrt{\pi} \Gamma(\frac{3}{2})}{8} \frac{F\left(\frac{3}{2}, \frac{3}{2}; \frac{3}{2}; \frac{\alpha_U^2}{(\alpha_1+\alpha_3+\alpha_U)(\alpha_2+\alpha_4+\alpha_U)}\right)}{(\alpha_1 + \alpha_3 + \alpha_U)^{\frac{3}{2}} (\alpha_2 + \alpha_4 + \alpha_U)^{\frac{3}{2}}}. \tag{113}$$

Now we can use that $F(a,b;b,z) = (1-z)^{-a}$ and that $\Gamma(\frac{3}{2}) = \frac{\sqrt{\pi}}{2}$. This finally yields

$$I_{\phi^4} = \frac{\pi}{16} [(\alpha_1 + \alpha_3 + \alpha_U)(\alpha_2 + \alpha_4 + \alpha_U) - \alpha_U^2]^{-\frac{3}{2}}. \tag{114}$$

Of course, this matrix element, containing only the angular momentum zero modes, could also have been obtained in simpler ways.

For the terms in Eq. (88) containing $F$ and $\phi^2$ we have that $l_1 = l_2$, $m_1 = -m_2 = m$ and $l_3 = l_4 = 0$. This leads to

$$I_{F\phi^2} = (-1)^m \frac{\pi(2l+1)!!}{2^{l+4}} \alpha_U^l [(\alpha_1 + \alpha_3 + \alpha_U)(\alpha_2 + \alpha_4 + \alpha_U) - \alpha_U^2]^{-(l+\frac{3}{2})}. \tag{115}$$

We find the same contribution, except for the $(-1)^m$ term, for the terms in Eq. (88) containing $G$ and $\phi^2$, in the case where $l_1 = l_4$, $m_1 = m_4 = m$ and $l_2 = l_3 = 0$. There are also $G$-$\phi^2$ contributions with $l_1 = l_3$, $m_1 = m_3 = m$ and $l_2 = l_4 = 0$, for which we find

$$I_{G\phi^2} = \frac{\pi(2l+1)!!}{2^{l+4}} (\alpha_2 + \alpha_4 + \alpha_U)^l [(\alpha_1 + \alpha_3 + \alpha_U)(\alpha_2 + \alpha_4 + \alpha_U) - \alpha_U^2]^{-(l+\frac{3}{2})}. \tag{116}$$

In case of the $F^2$ terms, we have that $l_1 = l_2 = l$, $m_1 = -m_2 = m$, $l_3 = l_4 = l'$ and $m_3 = -m_4 = m'$.

$$\sum_{m,m'} (-1)^{m+m'} I_{FF} = \frac{\sqrt{\pi}}{8} (2l+1)(2l'+1) \sum_L \frac{(2L+1)\left[\begin{pmatrix} l' & l & L \\ 0 & 0 & 0 \end{pmatrix}\right]^2 \Gamma(\frac{L+l+l'+3}{2})^2 \alpha_U^L}{\Gamma(L+\frac{3}{2})[(\alpha_1+\alpha_3+\alpha_U)(\alpha_2+\alpha_4+\alpha_U)]^{\frac{(l+l'+L+3)}{2}}}$$
$$F(\frac{L+l+l'+3}{2}, \frac{L+l+l'+3}{2}; L+\frac{3}{2}; \frac{\alpha_U^2}{(\alpha_1+\alpha_3+\alpha_U)(\alpha_2+\alpha_4+\alpha_U)}). \quad (117)$$

For the $G^2$ terms, we again find an identical result to Eq. (117) if $l_1 = l_4$, $m_1 = m_4 = m$, $l_2 = l_3$ and $m_2 = m_3 = m'$. If instead $l_1 = l_3$, $m_1 = m_3 = m$, $l_2 = l_4$ and $m_2 = m_4 = m'$, we find that

$$\sum_{m,m'} I_{GG} = \frac{\sqrt{\pi}}{8} (2l+1)(2l'+1) \frac{\Gamma(l+\frac{3}{2})\Gamma(l'+\frac{3}{2})}{\Gamma(\frac{3}{2})(\alpha_1+\alpha_3+\alpha_U)^{l+\frac{3}{2}}(\alpha_2+\alpha_4+\alpha_U)^{l'+\frac{3}{2}}}$$
$$F(l+\frac{3}{2}, l'+\frac{3}{2}; \frac{3}{2}; \frac{\alpha_U^2}{(\alpha_1+\alpha_3+\alpha_U)(\alpha_2+\alpha_4+\alpha_U)}). \quad (118)$$

Now we have all we need to find the expressions for $U_{00}$, $U_{10}$, $U'_{10}$ and $U_{11}$ from Eq. (37). For this, the integrals need to be multiplied by the prefactor $\frac{U}{2L_U^2}$. Furthermore, we need to add the prefactors following the Fourier transform to real space, see Eq. (96). Using this, we can fill in the above equations for the case of our analytical model:

$$U_{00} = \frac{U\pi L_U}{256\sigma_0^3} [L_U^2 + 4\sigma_0]^{-\frac{3}{2}}, \quad (119)$$

$$U_{10} = \frac{3U\pi L_U}{16\sigma_0^2} [L_U^2(\sigma_0+\sigma_1)^2 + 8\sigma_0\sigma_1^2 + 8\sigma_1\sigma_0^2]^{-\frac{5}{2}}, \quad (120)$$

$$U'_{10} = \frac{3U\pi L_U}{1024\sigma_0^{3/2}\sigma_1^{5/2}} (L_U^2+2\sigma_0)[L_U^2+2\sigma_0+2\sigma_1]^{-\frac{5}{2}}, \quad (121)$$

$$U_{11} = \frac{3U\pi L_U}{256\sigma_1^5} (L_U^2+4\sigma_1)^{-7/2}[\sigma_1^2 + \frac{(L_U^2+2\sigma_1)^2)}{16}]. \quad (122)$$