# Peer review of "Phase diagram for strong-coupling Bose polarons"

_SciPost Physics_

## Round 1 · Referee Report · Anonymous (Referee 1) · 2023-9-15

Report

The authors present a variational study of the Bose polaron problem in the regime of strong boson-impurity interactions, where for sufficiently light impurities one expects the Efimov effect to take place and modify the properties of the polaron. The authors compare two different theoretical scenarios for the fate of the polaron as one enters the regime of strong interactions and show that both are possible under different conditions. They also present a toy model in the spirit of Landau paradigm of phase transitions to illustrate their results. The problem of the Bose polaron at strong coupling is particularly interesting because of the overall complexity and the lack of many analytical results, so overall I find the results to be interesting and important for the development of the field. However, there is a significant weakness in the author’s work that needs to be addressed.

One of the theoretical possibilities mentioned above is the instability of the polaron that results in its decay into the multi-particle bound cluster. This instability was discovered by the same authors in Ref.[70, 71]. The approach used in Ref.[70,71] is based on the expansion of the theory around uniform condensate using the corresponding Bogoliubov modes and then truncating the theory at the quadratic level (apart from the Lee-Low-Pines term). In the case of the infinitely heavy impurity, it has been shown that this method does not correctly predict the energy of the polaron. Please refer to Fig. 4 of Ref.[88] where the analytical result obtained within this method is compared to the QMC calculation and also to the Appendix of https://journals.aps.org/pra/abstract/10.1103/PhysRevA.106.033305 for a toy model that shows the failure of the method. Because of the failure of the method for the infinitely heavy impurity, which is a simpler problem than that studied by the authors, I find the results of Ref.[70,71] and the corresponding scenario for the instability of the polaron to be questionable. On the other hand, in the current manuscript the authors seem to study the full Hamiltonian Eq.(4) without making any approximations or truncations, and obtain the same instability picture for the polaron. In that regard, I want the authors to address the following questions:

1. How do the results of the original study compare to the results obtained using the new variational method?

2. How seemingly wrong approach of Ref.[70,71] is capable of capturing the same instability picture? Is it some sort of insensitivity of the Efimov physics to the approximations made within this model?

3. The derivation of the analytical model in section 6 also seems to be inspired by the methods used in Ref.[70,71,74], so the authors should explain why this model is adequate for describing the physics at hand. For example, Eq.(22) looks very similar to Eq.(12), which does not give the correct result in the regime of strong coupling.

Other small comments and questions:

1. Page 8. When the authors cite different methods, the coherent state ansatz covers both the result in Eq.(12) and the results obtained using the Gross-Pitaevskii equation (GPe) in Ref.[91-93, 103,104]. I think the authors should make the distinction between both approaches a bit clearer and maybe mention the GPe in the text.

2. Page 9. “We set L_g = L_U = a_B and consider a high density BEC, n_0 = 10^-5 L_g^3”. This should be n_0 = 10^-5 L_g^(-3) for dimensional reasons. Same typo in Fig.2.

3. Fig. 2. Does CS1 correspond to the Gross-Pitaevskii equation (GPe)? If so, when comparing the results of the GS and CS1 on page 10, could you comment on whether the discrepancy within 3% is because of the quadratic fluctuations on top of the GPe result? According to Ref.[92] and https://journals.aps.org/pra/abstract/10.1103/PhysRevA.106.033305, the GPe is applicable when the local value of the gas parameter at the center of the impurity is small. The value of the gas parameter in Fig.2 is 10^-5 very far from the impurity, so even the enhancement of the density by the factor of 1000 should still result in a small correction to the result at unitarity. In that respect is it fair to say that the GS ansatz effectively accounts for the fluctuations in the vicinity of the impurity?

4. Page 14. “energy should depend approximately linearly on the density”. Again, authors cite Eq.(12), which has been shown to be incorrect at strong coupling.

I like that the authors investigate the problem in different regimes and make a very detailed comparison between two variational methods and discuss the limitations of both. This is one of the strongest sides of the manuscript. I will give my recommendation for the publication once the above issues are addressed.
  • validity: -
  • significance: -
  • originality: -
  • clarity: -
  • formatting: -
  • grammar: -

Author:  Arthur Christianen  on 2023-10-01  [id 4020]

(in reply to Report 1 on 2023-09-15)
Category:
remark
reply to objection

We thank the referee for their careful assessment of the manuscript. Here we would like to clarify our perspective regarding the presumed "weakness" of our paper mentioned by the referee. The referee points out that the approach from our previous work in Ref. [70,71] does not correctly capture the result for the infinitely heavy impurity, as presented in Ref. [88] and also the appendix of the paper the referee is mentioning. However, our approach in Refs. [70,71] was never meant to describe the behavior of the polaron in this regime. Instead, the aim was to describe the behavior of the polaron for light impurities and in the limit of non-interacting or very weakly interacting bosons. Thus, while we agree with the assessment of the referee and the results of Ref. [88] concerning the infinite mass limit, this is just not the limit our works tried to solve. In this large mass limit, Efimov correlations and much of the underlying physics we are interested in here, are strongly suppressed.

Focusing on the limit of light impurities and in absence of interboson repulsion, we presume the referee agrees with us that the picture of the instability of Ref. [70,71] is valid. For example, we show in Ref. [70,71] that the Efimov effect for non-interacting bosons is fully recovered. The referee finds it "questionable" to which degree these results hold in presence of explicit interboson repulsion, and this is in fact one of the key questions that we aimed to answer with our current manuscript.

Indeed, so far there has been no theoretical work which unites into a single picture both the phenomena predicted for non-interacting bosons, and for interacting bosons and an infinitely heavy impurity. In the present work, we fill this gap by our systematic comparison of the Gaussian-state approach and the double-excitation approach. We find that for light impurities and modest interboson repulsion, the instability picture of Ref. [70,71] can persist, but for heavier impurities we find the crossover behavior from Ref. [88] and other references. The results we find are therefore consistent with all the references the referee is citing. Note that the Eq. (22) which the referee points out is merely an intermediate step in our calculation where we consider non-interacting bosons, and the agreement with Eq. (12) is therefore not surprising. Later in the derivation of the analytical model we include the interboson interactions and then we find a different result.

We hope that the referee views our work in a more favorable light after this explanation. We will explain our reasoning more clearly in the revised manuscript, and we will also address the referee’s other comments, for which we are thankful.

---

## Round 1 · Referee Report · Anonymous (Referee 2) · 2023-10-24

Report

The paper considers the Bose polaron problem, where there is an impurity interacting with a Bose-Einstein condensate. This scenario has been attracting a lot of attention recently and there has been much debate about how to describe it theoretically. The challenge is to include the low-energy mean-field attributes of the condensate, such as the healing length, while at the same time describing the correlations that are relevant at short length scales close to the impurity, e.g., Efimov physics. Previous approaches based on coherent states have captured the mean-field physics, but have generally failed to account for correlations at short distances. So far, the current authors in Ref. [70] have successfully included Efimov trimers in a coherent-state-type approach but have not properly accounted for boson-boson repulsion at short distances, while Drescher et al in Ref [103] have included repulsive boson-boson correlations but not Effimov physics. The current work aims to address this gap.

Overall, I think the authors are addressing an important problem that should be of interest to a broad range of researchers. I also like the idea of using a quantum chemistry-inspired computational technique to tackle the problem and the fact that they’ve attempted to compare their Gaussian-state ansatz with the very different (i.e., non-coherent state) Chevy-type ansatz with two excitations of the BEC. I just had a few comments that the authors should address before publication:

  • In the discussion about cooperative binding on page 7, Ref [70] is cited for bound states beyond three particles, yet Ref. [70] only describes the trimer exactly, not higher body states. It would be more appropriate to cite Ref [69], which solves the 4-body problem exactly, and Blume PRA 99, 013613 (2019), which considers bound states up to 6 particles.

  • On page 8, where the double-excitation ansatz is discussed, it is stated: “… a similar method with an intrinsically limited number of excitations [68]” I found this a bit cryptic since Ref [68] considers the Bose polaron in the high-temperature limit where there is no BEC, which is somewhat different from the ground-state problem considered in this paper and others. I think this should be made clearer.

  • I must admit I found the treatment of the interboson repulsion rather confusing, because of how the authors have approximately treated it by inserting the Born approximation in various terms. It also muddies the comparison between the two different variational approaches, since it no longer involves just an expectation value of a Hamiltonian. How does the comparison look if the “bare” interactions are used?

  • I would be very interested to see how this Gaussian-state ansatz compares with the results of QMC, e.g., for the equal-mass case.

  • The predicted phase diagram features instabilities towards cluster states that involve multiple bosons bound to the impurity. However, the Gaussian-state ansatz only includes enough correlations to capture the Efimov trimer, not higher body bound states. The authors state that this is unlikely to be important for experimental observables, but surely this can affect the predicted instabilities in the phase diagram? E.g., if the actual bound states are much deeper in energy and therefore very far detuned from the polaron.

  • Do the authors have a sense of how three-body losses might impact the results?

  • There is the suggestion that the double-excitation ansatz does not properly describe the weak-coupling Bose polaron (see discussion on page 21) which I found puzzling since it has been shown to reproduce perturbation theory up to third order (see Ref. [67]), both for the quasiparticle residue and for the energy. As far as I know, this has not been demonstrated for the Gaussian-state ansatz. Is this something the authors have considered? I would suggest that this statement be modified.

  • validity: -
  • significance: -
  • originality: -
  • clarity: -
  • formatting: -
  • grammar: -

Author:  Arthur Christianen  on 2024-01-17  [id 4256]

(in reply to Report 2 on 2023-10-24)
Category:
answer to question

We thank the referee for their appreciation of our work and for their detailed and useful comments on our manuscript. Below we address their comments individually.

Referee: In the discussion about cooperative binding on page 7, Ref [70] is cited for bound states beyond three particles, yet Ref. [70] only describes the trimer exactly, not higher body states. It would be more appropriate to cite Ref [69], which solves the 4-body problem exactly, and Blume PRA 99, 013613 (2019), which considers bound states up to 6 particles.

Reply: We thank the referee for pointing this out, and we have added the other references at this point.

Referee: On page 8, where the double-excitation ansatz is discussed, it is stated:
“… a similar method with an intrinsically limited number of excitations [68]”
I found this a bit cryptic since Ref [68] considers the Bose polaron in the high-temperature limit where there is no BEC, which is somewhat different from the ground-state problem considered in this paper and others. I think this should be made clearer.

Reply: We agree with the referee that we should have pointed out more clearly the difference between this approach and the variational approaches we discuss in the same context, we have added the following sentence to the manuscript:
“This is similar when the virial expansion is performed, when truncating on the level of few excitations [68,85]”

Referee: I must admit I found the treatment of the interboson repulsion rather confusing, because of how the authors have approximately treated it by inserting the Born approximation in various terms. It also muddies the comparison between the two different variational approaches, since it no longer involves just an expectation value of a Hamiltonian. How does the comparison look if the “bare” interactions are used?

Reply: We understand the referee’s sentiment, and that is why we devoted Section 4 to benchmarking our treatment of the interboson repulsion, where the final Gaussian-state result behaves as expected. However, to further clarify the underlying reasoning behind this approach, we have substantially expanded Appendix C. In particular, we have included a novel figure, where we show that inserting the Born coupling constant is indeed necessary to get good results, in particular for the shape of the polaron cloud. We also compare in more detail the Gaussian-state result with the double-excitation result in the intermediate-coupling regime at an attractive scattering length. We find that there is a small but clear difference between the two methods, due to the difficulty of including the proper cubic interboson repulsion term in the double-excitation Ansatz. However, we do not expect this effect to play a substantial role for our main results in the strong-coupling regime.

Referee: I would be very interested to see how this Gaussian-state ansatz compares with the results of QMC, e.g., for the equal-mass case.

Reply: In response we have added the following paragraph to the main text at the end of the results section:
“Quantum Monte Carlo calculations have previously been performed mostly for equal mass or heavy impurities in presence of significant repulsion [90, 91, 97], and here good agreement was found with results from the double [90]-or triple-excitation [69] Ansatz. In this parameter regime these variational approaches outperform the Gaussian-state approach. In the regime of significantly lighter impurities and weak repulsion, where Gaussian states perform better, so far no Quantum Monte Carlo calculations were performed. This would be a very interesting avenue of research. “

Referee: The predicted phase diagram features instabilities towards cluster states that involve multiple bosons bound to the impurity. However, the Gaussian-state ansatz only includes enough correlations to capture the Efimov trimer, not higher body bound states. The authors state that this is unlikely to be important for experimental observables, but surely this can affect the predicted instabilities in the phase diagram? E.g., if the actual bound states are much deeper in energy and therefore very far detuned from the polaron.

Reply: The referee makes a good point, which was also discussed in more detail in our previous work. We have added the following paragraph in the Discussion part of the Results section:
“The Gaussian-state approach includes only pairwise interboson correlations and it is therefore important to discuss what would happen if higher-order correlations would be included. The polaronic instability is a cascade process, and these higher-order correlations will affect both the onset and the endpoint of the cascade. For the Gaussian-state Ansatz the onset of the instability can be viewed as a many-body shifted three-body Efimov resonance [70,71]. The polaron first decays into a few-body bound state before decaying into larger clusters. If up to n-body correlations would be included, then the instability would instead occur at a many-body shifted n-body Efimov resonance. However, the relevant timescale for these higher-order processes might be too slow to observe. If the order of the included correlations is as high as the number of particles of the cluster which first appears from the continuum, then one would find no instability any more, but a sharp crossover. However, for large clusters we expect that the avoided crossing in the spectrum between the polaron and the cluster would be extremely narrow.”

Referee: Do the authors have a sense of how three-body losses might impact the results?
Reply: We have already mentioned in the discussion part of the results section that three-body recombination would lead to a very short lifetime for large bound clusters, making them difficult to experimentally observe. However, to emphasize this more, we have added to the conclusion:
“From dynamical calculations also the polaron spectral function can be extracted, which can then be compared to experimental data. For a good comparison, it will also be important to assess what is the role of recombination. This will lead to spectral broadening especially for many-body bound states, and this might therefore make it difficult to experimentally observe such states.”

Referee: There is the suggestion that the double-excitation ansatz does not properly describe the weak-coupling Bose polaron (see discussion on page 21) which I found puzzling since it has been shown to reproduce perturbation theory up to third order (see Ref. [67]), both for the quasiparticle residue and for the energy. As far as I know, this has not been demonstrated for the Gaussian-state ansatz. Is this something the authors have considered? I would suggest that this statement be modified.

Reply: We thank the referee for pointing this out and we realize we should have made this statement more precise. This also relates to the previous comment about the description of the interboson repulsion. We have changed this statement in the main text to:
“Another shortcoming of our implementation of the double-excitation Ansatz is its description of the intermediate-coupling Bose polaron in presence of explicit interboson repulsion, which especially gives problems in the cubic term (see App. C). When the repulsion is described on the Bogoliubov level there is no problem, and the results from the double-excitation Ansatz agree with perturbation theory up to third order [67]. However, when explicitly taking into account the repulsion beyond the Bogoliubov approximation, consistently describing the repulsion with the double-excitation Ansatz would require an accurate description of the interactions in the background BEC. In this case it is more natural to describe the polaron cloud on the same footing as the background BEC, as done using a coherent- or Gaussian-state Ansatz.”

---

## Editorial Decision

resubmitted